# Sirt6 reprograms myofibers to oxidative type through CREB-dependent Sox6 suppression

Mi-Young Song[1,4], Chang Yeob Han[2,4], Young Jae Moon[1], Ju Hyung Lee[3], Eun Ju Bae[2✉] & Byung-Hyun Park [1✉]

Expanding the exercise capacity of skeletal muscle is an emerging strategy to combat obesity-related metabolic diseases and this can be achieved by shifting skeletal muscle fibers toward slow-twitch oxidative type. Here, we report that Sirt6, an anti-aging histone deacetylase, is critical in regulating myofiber configuration toward oxidative type and that Sirt6 activator can be an exercise mimetic. Genetic inactivation of Sirt6 in skeletal muscle reduced while its transgenic overexpression increased mitochondrial oxidative capacity and exercise performance in mice. Mechanistically, we show that Sirt6 downregulated Sox6, a key repressor of slow fiber specific gene, by increasing the transcription of CREB. Sirt6 expression is elevated in chronically exercised humans, and mice treated with an activator of Sirt6 showed an increase in exercise endurance as compared to exercise-trained controls. Thus, the current study identifies Sirt6 as a molecular target for reprogramming myofiber composition toward the oxidative type and for improving muscle performance.

[1] Department of Biochemistry and Molecular Biology, Chonbuk National University Medical School, Jeonju 54896, Republic of Korea. [2] College of Pharmacy, Chonbuk National University, Jeonju 54896, Republic of Korea. [3] Department of Preventive Medicine, Chonbuk National University Medical School, Jeonju 54896, Republic of Korea. [4]These authors contributed equally: Mi-Young Song, Chang Yeob Han. ✉email: ejbae7@jbnu.ac.kr; bhpark@jbnu.ac.kr

Skeletal muscle is composed of two fiber types that are dependent upon the different metabolic pathway for ATP production and the corresponding contraction pattern: type I (red-colored, slow-twitch) and type II (white-colored, fast-twitch) fibers. The former is rich in mitochondria and suited to performing long-lasting contraction, while the latter depends mainly on anaerobic glycolysis for its energy and is thus ideal for high-intensity contraction[1]. In mammalian skeletal muscles, multiple fiber types are generally intermingled within a single muscle, and each muscle has varying proportions of fiber types. Whereas soleus muscle is predominantly composed of type I oxidative fibers, extensor digitorum longus (EDL) muscle is rich in type II glycolytic fibers[2]. These proportions are plastic, and muscles have the ability to remodel their fiber types to adapt to different pathophysiological conditions. Whereas obesity and diabetes involve a shift toward type II fibers, with an inverse correlation between the proportion of type I fiber and intramuscular fat content[3], endurance exercise training increases the proportion of oxidative fibers[4]. Despite the importance of muscle fiber-type determination, however, the molecular basis of the regulatory process remains little understood.

Studies have identified several transcriptional regulators that control muscle fiber differentiation and fiber-type switching[5–7]. Peroxisome proliferator-activated receptor (PPAR)-γ coactivator-1α (PGC-1α) is an activating regulator for oxidative muscle remodeling, which involves elevated mitochondrial biogenesis and a shift from glucose to fatty acid as the main energy source[8,9]. Specifically, the ectopic expression of PGC-1α in skeletal muscle increases the proportion of oxidative fibers[10,11], whereas mice lacking PGC-1α exhibit a decline in oxidative metabolism in muscle and exercise capacity[12]. Conversely, the transcription factor SRY-box transcription factor 6 (Sox6) suppresses mitochondrial oxidative capacity and exercise performance[13,14]. Mechanistically, Sox6 cooperates with nuclear factor I X as a transcriptional repressor of the type I fiber-specific gene *Myh7*[15]. PGC-1α and Sox6 thus play opposing roles in the metabolic and contractile properties of skeletal muscle. The upstream regulatory mechanisms that determine these factors in fiber-type switching remain little known. Since aging is the main cause of fast-to-slow fiber-type shift, in addition to loss of muscle mass[16], we hypothesized that aging-related molecules may be associated with fiber-type specification in skeletal muscle.

Sirtuins (Sirt1-Sirt7) have been implicated in aging and aging-related diseases by deacetylating histone and non-histone proteins[17–19]. In particular, Sirt6 has emerged as a therapeutic target for the treatment of metabolic diseases[20]. A recent study with a cardiac/skeletal muscle-specific *Sirt6* knockout (KO) mouse (*Sirt6flox/flox*;*Ckmm-Cre*) model has suggested that while Sirt6 is critical for glucose homeostasis and insulin sensitivity it does not affect PGC-1α transcription or the associated mitochondrial biogenesis[21]. However, Samant et al.[22] have shown degenerating mitochondria with autophagic vacuoles in *Sirt6* deficient heart tissues. Thus, the precise roles of Sirt6 in skeletal muscle biology specifically in myofiber switching have thus far not been clearly elucidated, and its effect specifically on myofiber switching has not been previously explored. To investigate the role of Sirt6 in muscle fiber-type specification and skeletal muscle adaptations to exercise, we used skeletal muscle-specific *Sirt6* KO (*Sirt6flox/flox*;*Myl1-Cre*) and global *Sirt6* transgenic (Tg) mice or AAV9-mediated overexpression of Sirt6 in muscle, with a particular focus on mitochondrial biogenesis. We also treated mice with a small molecule Sirt6 activator to confirm and evaluate its value as a therapeutic target. Our findings provide clear evidence that activation of Sirt6 deacetylase offers an effective way of inducing myofiber switching to oxidative phenotype, thereby enhancing exercise performance in skeletal muscle.

## Results

**Sirt6 ablation in skeletal muscle leads to a decrease in exercise performance.** We first screened the sirtuin family members using the human GEO database to identify a key molecule for muscle fiber-type determination and exercise capacity. The transcript level of *SIRT6* was the most upregulated in the skeletal muscle of individuals after exercise compared to sedentary subjects (Fig. 1a). This significant increase in Sirt6 protein was also confirmed in exercise-trained mice and accompanied a decrease in H3K9 acetylation as an indicator of Sirt6 activity (Fig. 1b). Sirt6 induction may be mediated by endocrine/paracrine signaling and/or excitation-transcription coupling. To explore this possibility, we tested whether Sirt6 expression is changed by the exercise associated myokine interleukin (IL)-6[23] and electrical muscle stimulation. Treatment of C2C12 myotubes with recombinant IL-6 (5-50 ng/ml) raised Sirt6 protein expression in a concentration-dependent manner (Supplementary Fig. 1a). Electrical muscle stimulation also increased Sirt6 expression in mice (Supplementary Fig. 1b), suggesting that multiple stimuli may coordinately contribute to Sirt6 upregulation in exercise conditions.

To further explore the physiological role of Sirt6 in skeletal muscle, we generated skeletal muscle-specific *Sirt6* KO mice by crossing *Sirt6flox/flox* mice with mice expressing Cre recombinase under the control of the myosin light chain 1 promoter (*Myl1-Cre*), which is mainly active in postmitotic type II myofibers[24,25] (Supplementary Fig. 2a). To visualize the skeletal muscle tissues in which Cre-mediated recombination occurred, we mated *Myl1-Cre* mice with *R26R* reporter mice. X-gal staining revealed strong recombination in the quadriceps femoris (QF), gastrocnemius (GAS), tibialis anterior (TA), and soleus (SOL) muscles (Supplementary Fig. 2b). Frozen section from GAS muscle tissue showed a specific β-gal expression in fast muscle cells (Supplementary Fig. 2c). Western blot analysis confirmed the specific ablation of Sirt6 in skeletal muscle tissues (Supplementary Fig. 2d). *Sirt6* KO mice had a phenotypically normal gross appearance at birth and showed similar basic metabolic parameters (i.e., body weight, lean body mass, epididymal adipose tissue weight, and amount of food intake) as their wild type (WT) littermates (Supplementary Fig. 3a–d).

We subjected the *Sirt6* KO and WT mice to involuntary exercise training on a treadmill for one month. The *Sirt6* KO mice exhibited significantly lower grip strength compared to the WT mice (Fig. 1c). Similarly, endurance capacity—shown as average running time, running distance, and run time to exhaustion—was significantly lower in the *Sirt6* KO mice (Fig. 1d–f). Intriguingly, indirect calorimetric analysis of energy metabolism revealed a significant increase in RER ($VCO_2/VO_2$) in the *Sirt6* KO mice compared to the WT mice (Fig. 1g), indicating a switch in mitochondrial substrate preference from fatty acid to glucose. All of these phenotypes were changed in opposite ways in the *Sirt6* Tg mice (Fig. 1h–k).

To further evaluate the effect of *Sirt6* deficiency on muscle function, we performed an ex vivo study to measure isometric force and fatigue sensitivity. The results indicated that GAS muscles from *Sirt6* KO mice exhibit reduced tetanic contraction and are more susceptible to fatigue than those from WT mice, though there was no significant difference in the twitch contraction between genotypes (Supplementary Fig. 4a–d).

**Sirt6 ablation decreases oxidative myofiber relative to glycolytic myofiber.** In general, impaired exercise endurance is associated with a decrease in the proportion of oxidative fibers[1]. Accordingly, we next examined the muscle fiber-type composition and cross-sectional areas of fiber types in several hind limb muscles. Upon visual inspection, each of the different types of

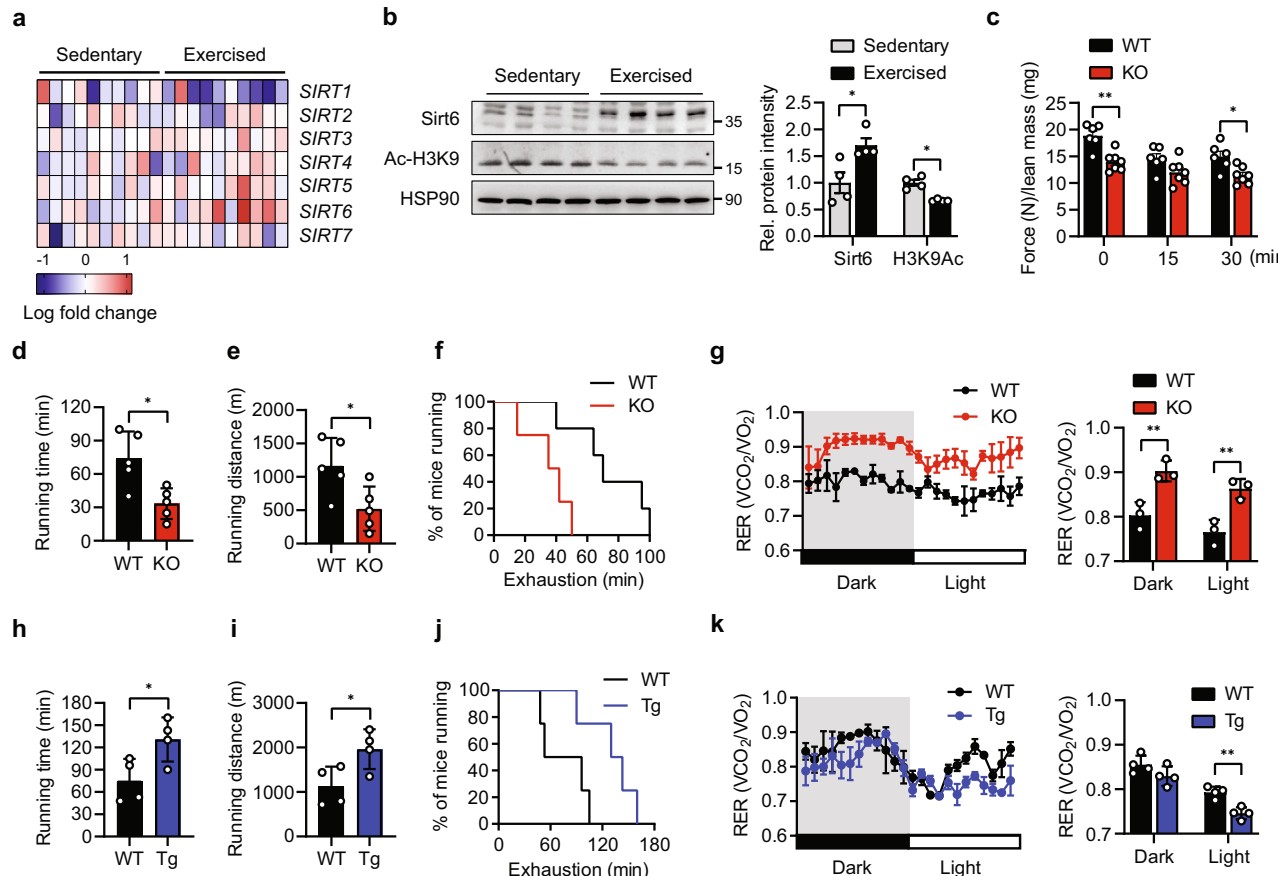

**Fig. 1 Alteration of exercise performance in *Sirt6* KO and *Sirt6* Tg mice. a** Heatmap visualizing expression of sirtuin family members in sedentary or exercise-trained human subjects (extracted from GSE9103) ($n = 10$). **b** Representative Western blot images. Expression of Sirt6 in gastrocnemius muscle was compared in sedentary or exercise-trained mice ($n = 4$). **c** Forelimb grip strengths were measured at 15 min intervals and normalized against lean body mass ($n = 6$ for WT, $n = 7$ for KO). **d–f**, **h–j** Treadmill endurance performance after four weeks of treadmill exercise training. Running time (**d**, **h**), distance (**e**, **i**), and running population plotted against time to exhaustion (**f**, **j**) in KO or Tg mice, respectively ($n = 5$ for (**d–f**), $n = 4$ for (**h–j**)). **g**, **k** Twenty-four hour light and dark cycle respiratory exchange ratio (RER) were measured by indirect colorimetry ($n = 3$ for (**g**), $n = 4$ for (**k**)). (**d–g**) WT and *Sirt6* KO mice, (**h–k**) WT and *Sirt6* Tg mice. Values are mean ± SEM. Data are representative of at least three independent experiments. Unpaired two-tailed *t* test between two groups was conducted for statistical analyses (**b–e**, **g–i**, **k**). *$p < 0.05$ and **$p < 0.01$. Source data are provided as a Source Data file.

muscles from the *Sirt6* KO mice were less red in color compared to those from the WT mice (Fig. 2a). Muscle fiber-type assessment of GAS muscles by immunofluorescent staining for myosin heavy chain (MyHC) isoforms revealed that *Sirt6* ablation led to decreased oxidative fiber density and increased glycolytic fiber size in comparison with those of WT mice (Fig. 2b and Supplementary Fig. 5a). In keeping with this, whereas the mRNA expression of slow fiber-specific genes such as MyHC-I (*Myh7*), MyHC-IIa (*Myh2*), troponin I (*Tnni1*), troponin C (*Tnnc1*), and troponin T1 (*Tnnt1*) were significantly downregulated, the fast fiber-specific gene MyHC-IIb (*Myh4*) was dramatically upregulated in GAS muscle from the *Sirt6* KO mice (Fig. 2c). When muscle sections were immunostained for succinate dehydrogenase (SDH), a marker of mitochondrial activity, less than 50% of fibers were SDH positive in the muscle of the *Sirt6* KO mice, whereas over 70% of myofibers were SDH positive in the WT mice (Fig. 2d). Similar observations were made in the case of EDL and SOL muscles of *Sirt6* KO mice with regard to the proportion of myofibers and the expression of fiber-type-specific genes (a much weaker effect was observed in SOL muscles presumably due to the predominant action of *Myl1-Cre* in fast muscles) (Supplementary Fig. 6a–c), whereas the opposite results were obtained from GAS and EDL muscles from whole body *Sirt6* Tg mice (Fig. 2e–h and Supplementary Figs. 5b, 7).

Moreover, intramuscular overexpression of Sirt6 using an AAV9 delivery system enhanced oxidative fiber-type composition and exercise performance (Supplementary Fig. 8a–e). Overall, these findings suggest that Sirt6 mediates the shift of myofibers towards an oxidative phenotype by increasing the proportion of slow twitch myofibers.

Since *Myl1-Cre* action starts embryonically during myoblast differentiation[25,26], we further inquired whether the effect of Sirt6 deletion on myofiber composition is a consequence of developmental reprogramming or arises as a result of conversion between different types of mature myofibers. We introduced Sirt6 siRNA into C2C12 cells at various stages of myogenesis. Regardless of transfection time (at day 1 or 5 after differentiation), Sirt6 silencing decreased the mRNA levels of slow fiber genes but increased the mRNA levels of fast fiber gene in C2C12 myotubes (Supplementary Fig. 9a, b). These results were verified in tests with electrical stimulation (Supplementary Fig. 9c, d). In sum, our results indicate that Sirt6 may act as a regulator of muscle fiber-type composition during myogenesis and in mature myofibers for conversion.

**Sirt6 ablation reduces mitochondrial content and oxidative capacity.** Electron microscopic analysis of GAS muscle revealed that the *Sirt6* KO mice showed slight but nevertheless

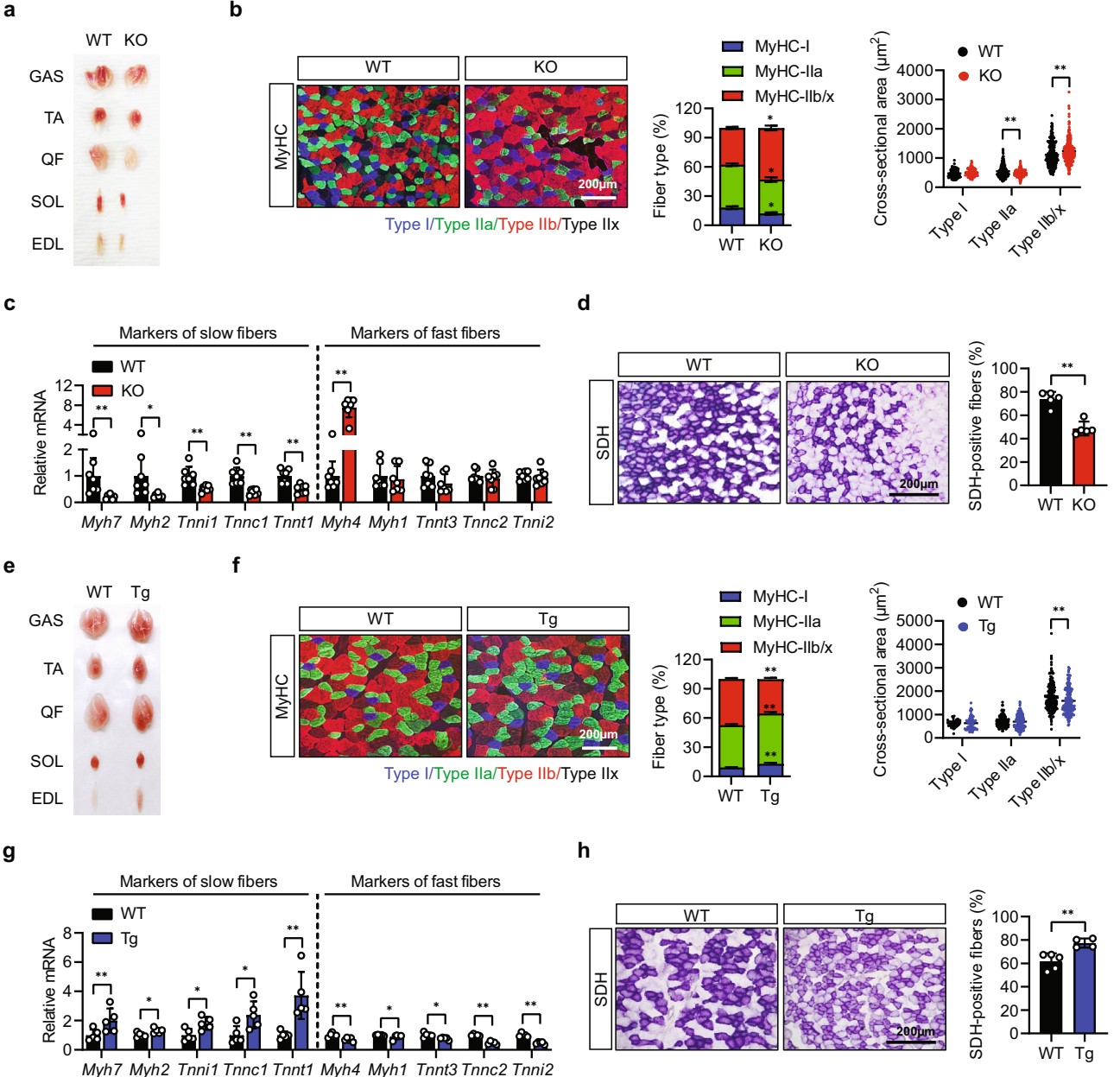

**Fig. 2 Alteration of myofiber composition in gastrocnemius muscle of *Sirt6* KO and *Sirt6* Tg mice at basal condition. a, e** Representative photographs showing skeletal muscles from 20-week old WT and *Sirt6* KO mice or 15-week old WT and *Sirt6* Tg mice. **b, f** Representative immunofluorescence staining for MyHC-I, MyHC-IIa, and MyHC-IIb/x. Composition and cross-sectional area (CSA) of each myofiber were quantified ($n = 6$ for (**b**), $n = 4$–6 for (**f**)). Bar=200 μm. **c, g** Expression of markers of slow and fast fibers was compared by qPCR ($n = 7$ for (**c**), $n = 5$ for (**g**)). **d, h** Representative succinate dehydrogenase (SDH) staining and quantification of SDH-positive fibers ($n = 5$ for (**d**), $n = 4$–5 for (**h**)). Bar=200 μm. (**a–d**) WT and *Sirt6* KO mice, (**e–h**) WT and *Sirt6* Tg mice. Values are mean ± SEM. Data are representative of at least three independent experiments. Unpaired two-tailed *t* test between two groups was conducted for statistical analyses (**b–d**, **f–h**). *$p < 0.05$ and **$p < 0.01$. Source data are provided as a Source Data file.

significantly less mitochondria relative to the WT mice (Fig. 3a, b). Of note, abnormal mitochondria with the destruction of cristae had markedly increased in the *Sirt6* KO mice. These findings correlate well with the decreased mitochondrial DNA content and suppressed gene expression with regard to mitochondrial biogenesis and oxidative phosphorylation in muscle tissues from *Sirt6* KO mice (Fig. 3c–e). Conversely, mitochondrial content and gene expressions were elevated in the *Sirt6* Tg mice (Supplementary Fig. 10a, b).

Because mitochondrial content and quality control are affected by alterations in mitochondrial dynamics[27], we measured genes associated with mitochondrial fusion-fission proteins in the GAS

muscles of *Sirt6* KO and WT mice. In contrast with the marked suppression of genes related to mitochondrial biogenesis, mRNA and protein levels of mitochondrial fusion-fission genes (i.e., OPA1, Mfn1, Drp1, and Fis1) remained unaffected in KO relative to WT mice (Supplementary Fig. 11a, b), supporting the notion that the effect of Sirt6 on mitochondrial contents primarily arises as a result of the regulation of mitochondrial biogenesis rather than the control of dynamics.

To determine whether the decrease in mitochondria numbers in the *Sirt6* KO mice was associated with decreased respiratory function, we performed a Seahorse XF Mito Stress test in myofibers isolated from GAS muscles. In conjunction with

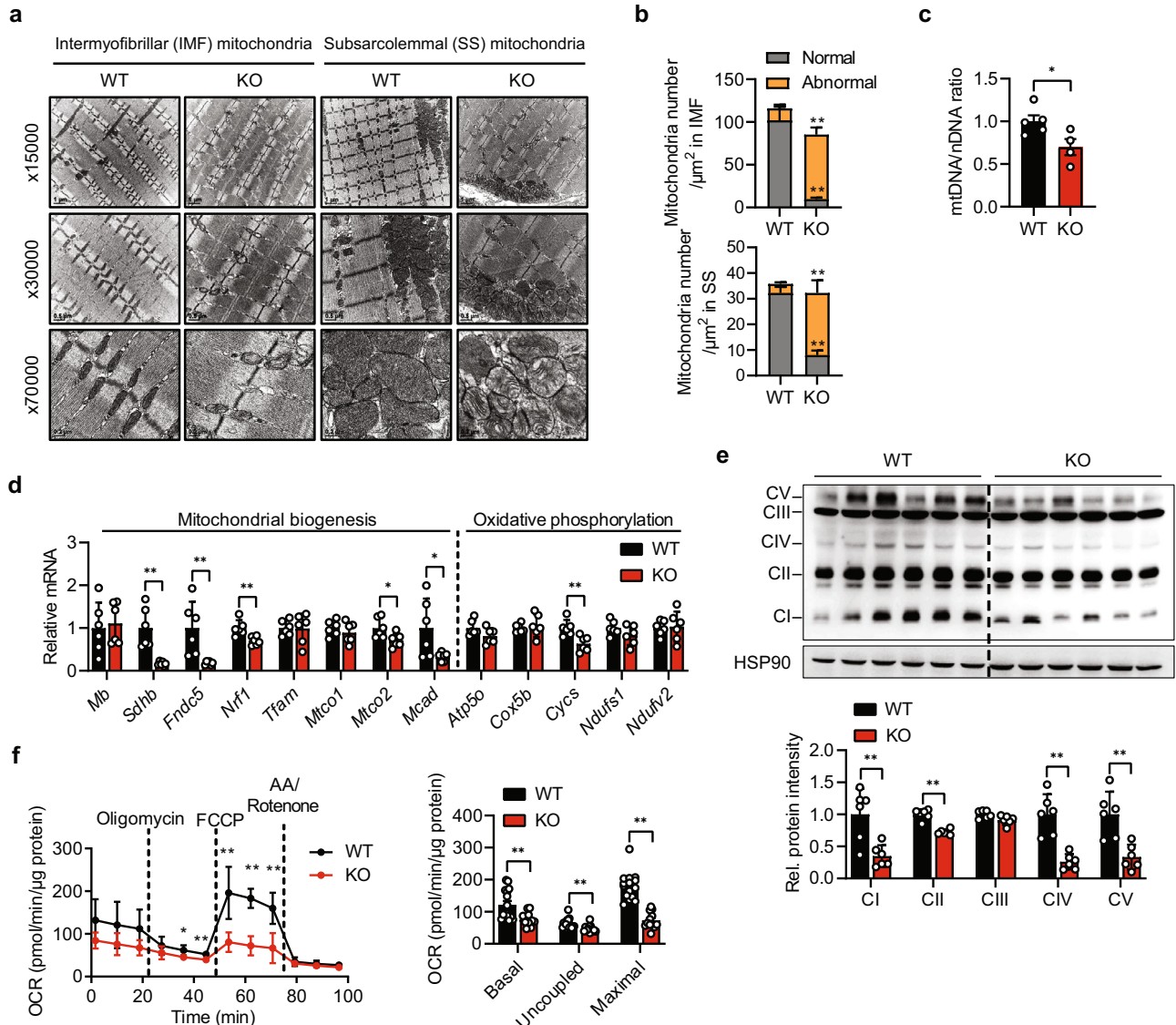

**Fig. 3 Decrease in oxidative capacity in *Sirt6* KO mice at basal condition. a** Representative transmission electron micrograph of gastrocnemius muscles (GAS) from WT and *Sirt6* KO mice. **b** Numbers of mitochondria with normal or abnormal morphology were counted from images in (**a**) (*n* = 4). **c** Mitochondrial DNA (mtDNA) was quantified by qPCR using nuclear DNA (nDNA) as a standard (*n* = 5 for WT, *n* = 4 KO). **d** qPCR analysis of genes related to mitochondrial biogenesis and oxidative phosphorylation (OxPhos) in GAS (*n* = 6). Expression of each gene was normalized with housekeeping *Gapdh* whereas expression of mitochondrial genome-encoded genes *Mtco1* and *Mtco2* was normalized with *16S rRNA*. **e** Representative Western blot analysis of OxPhos complex (*n* = 6). **f** Oxygen consumption rate (OCR) was measured using Seahorse XF analyzer in myofibers isolated from GAS. Basal respiration, respiration related to ATP production (uncoupled, difference between OCR before and after oligomycin injection), and maximal respiration (difference between OCR after FCCP and antimycin A (AA)/rotenone injection) were determined (*n* = 15 for WT, *n* = 12 for KO). Values are mean ± SEM. Data are representative of at least three independent experiments. Unpaired two-tailed *t* test between two groups was conducted for statistical analyses (**b**–**f**). *$p < 0.05$ and **$p < 0.01$. Source data are provided as a Source Data file.

reduced mitochondrial abundance, we observed suppression of basal, ATP-linked, and maximal respiration in the *Sirt6* KO mice (Fig. 3f), indicating that *Sirt6* ablation impairs the ability to meet cellular energy demands through oxidative phosphorylation.

**Sirt6 ablation suppresses CREB transcription and its downstream genes.** To trace the molecular mechanisms by which Sirt6 modulates fiber-type specification and mitochondrial biogenesis, we performed RNA-seq analysis using GAS muscles from *Sirt6* KO mice. Upregulated (red colored) or downregulated (blue colored) genes in the muscles of the *Sirt6* KO mice relative to those of the WT mice were demonstrated in a volcano plot (Fig. 4a). Among the top 20 downregulated genes, multiple target

genes downstream of cAMP response element-binding protein (CREB) were found, notably PGC-1α (*Ppargc1a*), Nor1 (*Nr4a3*), and Nur77 (*Nr4a1*), which are related to muscle fiber specification and mitochondrial respiration (Fig. 4b). qPCR analysis and Western blotting further confirmed changes in the expression of these genes (Fig. 4c, d).

Since the role of Sirt6 in CREB regulation has never been explored, we then investigated a possible relationship between Sirt6 and CREB. Analysis of the human GTEx database revealed a strong positive correlation between *SIRT6* and *CREB* in the skeletal muscle tissues of 564 individuals (Fig. 4e). Consistent with this, CREB mRNA and protein levels were markedly downregulated in *Sirt6* KO muscles (Fig. 4f, g), while markedly

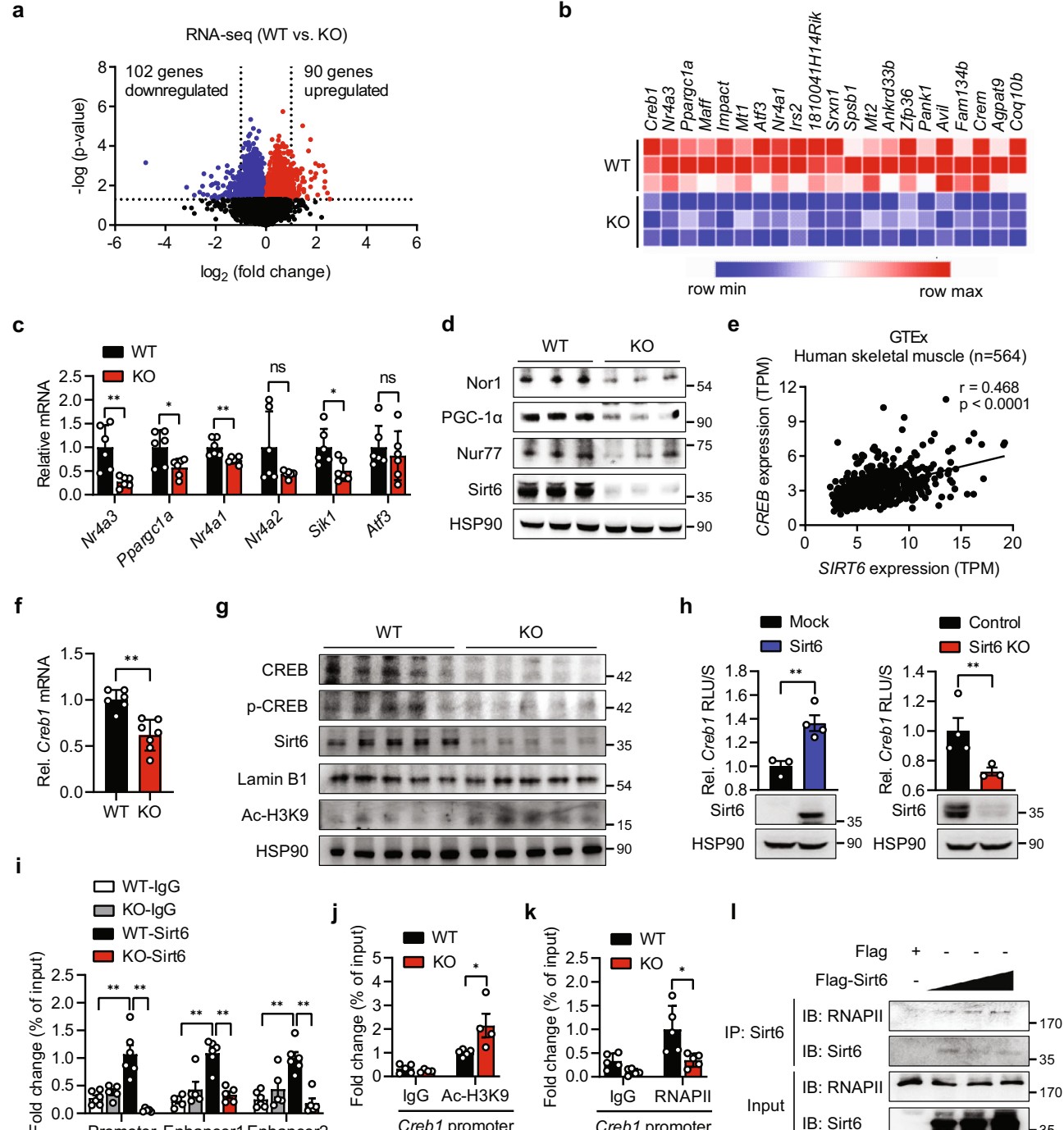

**Fig. 4 Decrease in CREB pathway in muscle of *Sirt6* KO mice. a** Volcano plot showing the $log_2$ fold-difference in mRNA expression in gastrocnemius muscles (GAS) of WT and *Sirt6* KO mice, assessed by StringTie analysis of RNA-seq data. Red and blue dots represent upregulated and downregulated DEGs with > 2-fold change and $p < 0.05$, respectively. Each sample was analyzed in triplicate. **b** Heat map of RNA-seq analysis of *Creb1* and top 20 downregulated genes. **c, d** qPCR and Western blotting analyses of CREB target genes in GAS of mice ($n = 6$). **e** Genotype tissue expression (GTEx) analysis of human skeletal muscle. Pearson correlation coefficient (*r*) between SIRT6 and CREB in human skeletal muscle were calculated. TPM, transcripts per million. **f, g** *Creb1* mRNA and levels of the indicated protein in GAS of mice ($n = 6$ for WT, $n = 7$ for KO). **h** *Creb1*-luciferase reporter activities in HEK293T cells overexpressing *Sirt6* or in cells with CRISPR/cas9 mediated *Sirt6* knockout (KO) were measured at 24 h post-transfection and expressed as the fold change relative to Mock or Control ($n = 3$ for Mock, $n = 4$ Sirt6, $n = 4$ for Control, $n = 3$ for Sirt6 KO). **i** Association of Sirt6 to *Creb1* promoter or enhancers was evaluated by ChIP-qPCR in GAS of WT and *Sirt6* KO mice ($n = 5$–6). **j, k** ChIP-qPCR for the association of Ac-H3K9 ($n = 5$ for WT, $n = 4$ for KO) (**j**) and RNA polymerase II (RNAPII) ($n = 5$) (**k**) to *Creb1* promoter in GAS of WT and Sirt6 KO mice. **l** HEK293T cells were transfected with increasing amount of *Sirt6* and the binding of Sirt6 to RNAPII was determined. Values are mean ± SEM. Data and all images are representative of at least three independent experiments. Negative binomial Wald Test using DESeq2 (**a**), unpaired two-tailed *t* test between two groups (**c, e, f, h, j, k**) and one-way ANOVA followed by Bonferroni's post hoc analysis (**i**) were conducted for statistical analyses. *$p < 0.05$ and **$p < 0.01$. Source data are provided as a Source data file.

upregulated in *Sirt6* Tg muscles (Supplementary Fig. 12a, b), suggesting the bona-fide effect of Sirt6 on the regulation of CREB.

The *Creb1*-luciferase reporter gene assay was next performed in HEK293T cells and the results were consistent; Sirt6 over-expression significantly induced the luciferase signal, while Sirt6 ablation attenuated it (Fig. 4h). These results confirmed the positive role of Sirt6 in *Creb1* transactivation. ChIP assays then revealed that Sirt6 is enriched in the proximal promoter and distal enhancer regions of the *Creb1* gene in the skeletal muscles of control mice, which was clearly diminished in *Sirt6* KO mice (Fig. 4i). Sirt6, as an anti-aging molecule, has previously been shown to be downregulated over ages in various tissues[28–30], observations that were confirmed by our results that Sir6 protein was markedly reduced in the muscles of 20-week-old mice relative to 4-week-old mice (Supplementary Fig. 13a). The association of Sirt6 in *Creb1* promoters and enhancers was significantly diminished in 20-week-old mice (Supplementary Fig. 13b), suggesting the Sirt6 regulation of CREB in pathophysiological conditions.

Consistent with the observed increase in H3K9 acetylation in *Sirt6* KO mice (Fig. 4g), Ac-H3K9 occupancy in the *Creb1* promoter had also increased (Fig. 4j) while recruitment of RNA polymerase II (RNAPII) to the *Creb1* promoter was significantly dampened (Fig. 4k). This suggests that the intriguing possibility of a deacetylase activity dependent recruitment of RNAPII by Sirt6. We additionally observed the direct interaction of Sirt6 and RNAPII by co-IP in HEK293 cells transfected with plasmid expressing *Sirt6* (Fig. 4l).

**Sirt6 induces oxidative fiber conversion through the CREB-Sox6 axis**. The balance between transcriptional activators and repressors is critical in determining the number of oxidative muscle fibers[5,31]. Since coactivator PGC-1α, which is downstream of CREB, was altered by Sirt6 deficiency (Fig. 4c, d), we checked the transcription repressors of oxidative fiber conversion in *Sirt6* KO mice. Results showed that *Sox6*, but not chromobox protein homolog 1 (*Cbx1*), trans-acting transcription factor 3 (*Sp3*), or purine rich element-binding protein B (*Purb*), was markedly increased in *Sirt6* KO muscle (Fig. 5a, b). Conversely, *Sirt6* Tg muscles showed decreased Sox6 expression (Fig. 5c, d). Moreover, Sox6 binding to the promoter of slow fiber-specific gene *Myh7* was significantly increased by *Sirt6* ablation (Fig. 5e), supporting the inference of a functional increase of Sox6 in the *Sirt6* KO mice. Since CREB is a negative regulator of Sox6[32] and given our prior observation that Sirt6 enhanced CREB expression (Fig. 4), we measured whether CREB occupancy on *Sox6* promoter was altered in *Sirt6* KO muscle. CREB enrichment on *Sox6* promoter was observed, which was decreased by *Sirt6* ablation (Fig. 5f), implying that Sirt6-CREB-Sox6 axis regulates the expression of slow fiber-specific genes.

The role of Sox6 in the downregulation of slow muscle fibers has been already identified[14]. To demonstrate the causal relationship of Sox6 to Sirt6-dependent muscle fiber-type changes, and because the AAV9 serotype has been identified as an effective means of targeting skeletal muscle[33], we silenced *Sox6* in skeletal muscle using AAV9-mediated shSox6 delivery to mice (Fig. 5g). Western blotting, histology and biochemical analysis confirmed that the skeletal muscle-selective depletion of Sox6 had no effect on the heart and kidney tissues (Supplementary Fig. 14a–e). The reduction by *Sirt6* ablation of slow myofibers and corresponding increase in fast myofibers was efficiently restored by *Sox6* knockdown, which was confirmed by visual examination, immunostaining of MyHC isoforms, and qPCR analysis of fiber-type-specific genes (Fig. 5h–j). In addition, the ablation of Sox6 restored the population of SDH-positive fibers in

*Sirt6* KO muscle (Fig. 5k). Strikingly, the decreased exercise performance in the *Sirt6* KO mice was completely reversed by *Sox6* silencing (Fig. 5l–n). To further demonstrate the role of the Sirt6-CREB axis as an upstream regulator for Sox6, we performed these experiments in CREB knockdown cells. The results indicated that the effect of Sirt6 in supporting slow muscle fiber was lost in the CREB knockdown C2C12 myotubes (Supplementary Fig. 15a, b). Altogether, these results clearly demonstrate that Sirt6 inhibits Sox6 expression through CREB activation, thereby contributing to myofiber switching to oxidative phenotype and enhanced exercise performance.

**Pharmacological Sirt6 activation enhances exercise performance**. To provide a clinical proof-of-concept, we treated WT and *Sirt6* KO mice with the Sirt6 specific activator MDL801[34] for 4 weeks as they were subjected to chronic treadmill exercise. The on-target specific functionality of Sirt6 activator MDL801 was confirmed by H3K9 deacetylation in the muscle tissues, which effect was completely gone in *Sirt6* KO muscles (Fig. 6a). Other deacetylases (Sirt1 and HDAC11) targeting H3K9 were not altered by treatment with MDL801 (Supplementary Fig. 16a). Consistent with the increase of RER ($VCO_2/VO_2$) and decrease of OCR caused by Sirt6 deficiency (Figs. 1g, 3f), lower RER and higher OCR were observed in mice and C2C12 cells that received MDL801 treatment (Supplementary Fig. 16b, c), confirming the role of Sirt6 activation in the shift towards oxidative phosphorylation. Notably, running time, running distance, and time to exhaustion had increased approximately 2-fold in MDL801-treated mice compared to vehicle (Fig. 6b–d), an effect that was not observed in Sirt6 KO mice. Consistent with the observation in *Sirt6* Tg (Supplementary Fig. 12), MDL801 treatment increased the expression of *Creb1* and its target genes, while decreasing *Sox6* in mice (Supplementary Fig. 16d). In addition, the effects of MDL801 on the expression of myofiber type-specific genes were blocked by CREB silencing in C2C12 myotubes (Supplementary Fig. 16e). In terms of functional consequences, we observed that mice treated with MDL801 showed increased slow fiber composition and size with a concomitant increase in mitochondrial oxidative capacity in their GAS and EDL muscles (Fig. 6e, f and Supplementary Fig. 17).

## Discussion

In the current study, we have identified a previously unrecognized function of Sirt6 as a molecular switch for reprogramming myofibers to the oxidative type and as an enhancer of exercise endurance (Fig. 6g). While skeletal muscle-specific *Sirt6* ablation attenuated slow fiber recruitment, *Sirt6* transgenic overexpression promoted it, which was reflected in increased mitochondrial content and oxidative capacity. Through transcriptome analysis combined with physiological data, we identified CREB as a target of Sirt6 in determining fast-to-slow fiber conversion. In line with previous findings that CREB plays a pivotal role in the metabolic function of skeletal muscle[35], slow-fiber activating molecules downstream of CREB, including PGC-1α and Nr4a3/1 (encoding Nor1 and Nur77, respectively), were markedly upregulated by Sirt6. Conversely, Sox6, a potent repressor of slow muscle identity, was inhibited by Sirt6 and revealed to be downstream of CREB, thus specifying the Sirt6-CREB-Sox6 axis as a pathway in determining myofiber type. Lastly, the present study shows that Sirt6 activation may offer a promising exercise mimetic therapy, a conclusion corroborated by the evidence of Sirt6 upregulation by exercise in humans and mice. Maintaining Sirt6 protein level or activity in skeletal muscle is therefore essential to switching to the slow myofiber type and enhancing exercise endurance.

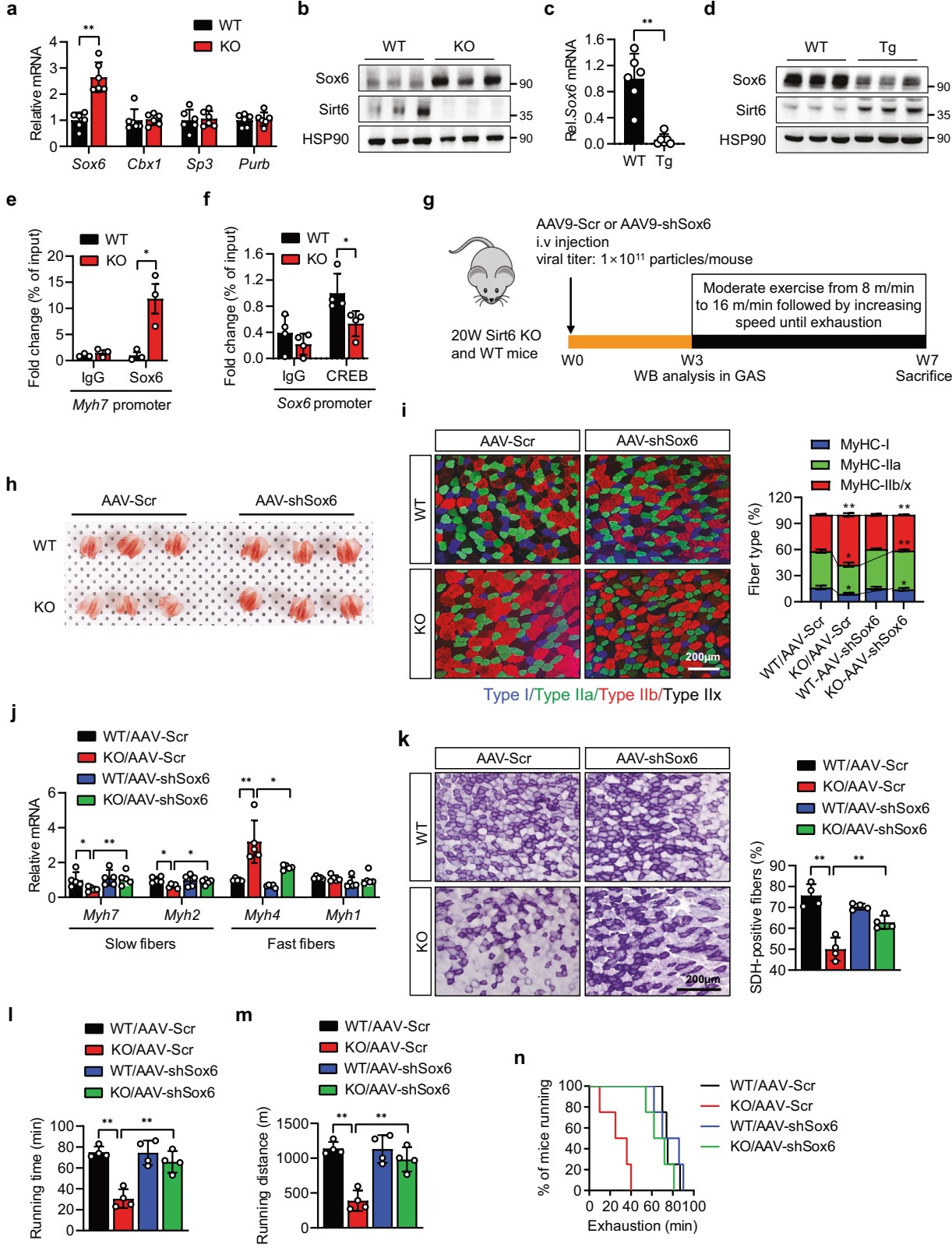

We utilized multiple animal models and in vivo gene delivery tools to manipulate Sirt6 gene expression specifically in skeletal muscle. Despite the varying degree of tissue specificity of Sirt6 gene manipulation, the results of the gene deletion (i.e., skeletal muscle-specific Sirt6 KO (*Myl1-Cre*) as well as AAV-shRNA mediated gene deletion) and overexpression (i.e., global Sirt6 Tg or intramuscular injection of AAV9-Sirt6) tests consistently affirmed that Sirt6 is a vital driver of oxidative, slow-twitch myofibers. Interestingly, although *Myl1* expression is specific to type II fibers and this was the case in our study- i.e., deletion

**Fig. 5 Reciprocal regulation between Sox6 and Sirt6 expression in muscle of *Sirt6* KO and Tg mice. a–d** mRNA and protein levels of transcriptional repressor genes of slow muscle fiber in gastrocnemius muscles (GAS) of *Sirt6* KO and Tg mice ($n = 6$). **e** ChIP-qPCR assay for Sox6 association to *Myh7* promoter ($n = 3$). **f** ChIP-qPCR assay for CREB association to *Sox6* promoter ($n = 4$). **g** Schematic for in vivo AAV-shSox6 study. Twenty-week-old WT and *Sirt6* KO mice were i.v. injected with AAV9 expressing a scrambled shRNA (AAV-Scr) or shSox6 (AAV-Sox6). Three weeks later, muscle tissues were collected to confirm Sox6 knockdown. Mice were further subjected to treadmill exercise for 4 weeks. **h** Gross observation of GAS. **i** Representative immunofluorescence staining for MyHC-I, MyHC-IIa, and MyHC-IIb/x. Composition of each myofiber was quantified ($n = 4$). Bar=200 µm. **j** qPCR for marker genes of slow and fast fibers ($n = 5$). **k** Representative succinate dehydrogenase (SDH) staining and quantification of SDH-positive fibers ($n = 4$). Bar = 200 µm. **l–n** Running time, running distance, and time to exhaustion after four weeks of treadmill exercise training ($n = 4$). Values are mean ± SEM. Data are representative of at least three independent experiments. Unpaired two-tailed *t* test between two groups (**a**, **c**, **f**) and one-way ANOVA followed by Bonferroni's post hoc analysis (**l**, **m**) were conducted for statistical analyses. *$p < 0.05$ and **$p < 0.01$. Source data are provided as a Source Data file.

efficiency of Sirt6 was higher in glycolytic muscles than in oxidative muscles (e.g., EDL vs. SOL), the effect of Sirt6 on muscle function and fiber-type specification was demonstrated consistently across all muscle type (GAS, EDL and SOL). As we focused for simplicity's sake on the distinction between slow and fast myofibers in interpreting our initial experimental results, additional studies will be required to determine the slow fiber selective role of Sirt6 and the degree to which Sirt6-CREB-Sox6 signaling participates in defining the complete spectrum of myofiber subtypes. In addition to the spatial limitation described above, our model *Myl1*-Cre-mediated Sirt6 KO mice raises the temporal limitation-developmentally early onset of *Sirt6* gene silencing-so it may be premature to draw a definitive conclusion the Sirt6 enhances exercise performance by facilitating the fast-to-slow fiber shift.

We demonstrated that mitochondrial biogenesis was significantly reduced in *Myl1*-Cre *Sirt6* KO mice, a result that conflicts with a previous report by Cui et al.[21], in which *Mck*-Cre-mediated *Sirt6* KO deletion did not show any change in mitochondrial biogenesis. The difference may be due to the timing of Sirt6 deletion in the development stages. While myosin light chain (*Myl1*) expression starts embryonically (E9), muscle creatine kinase (*Mck*) expression generally begins perinatally during the later stage of myotube maturation and myofiber formation (E15)[25,26]. Impairment of mitochondrial biogenesis in *Myl1*-Cre *Sirt6* KO mice may be the result of temporal effect of Sirt6 deletion, depending on the developmental stage of the modulated muscle.

Given that slow muscle fibers are rich in mitochondria and preferentially use oxidative phosphorylation as an energy source, it was not surprising to observe noticeable derangements in mitochondrial structure and respiratory function in the *Sirt6* KO mice. When observed by EM, around 80% of the mitochondria in *Sirt6* KO muscle displayed morphological defects characterized by disrupted cristae. These findings were consolidated by results that the mitochondria from GAS muscles of *Sirt6* KO mice were functionally impaired, evidenced by higher RER and lower oxygen consumption during exercise training. Although the CREB-Sox6 pathway was identified as a downstream target of Sirt6 in supporting myofiber specification and mitochondrial biogenesis, we propose that PGC-1α, a key player in mitochondrial biogenesis in skeletal muscle[8,9], is an additional link in mediating the benefits that Sirt6 provides to muscle. It was previously reported that PGC-1α is regulated by several upstream molecules including Sirt6[36], Sirt1, and AMP-activated protein kinase (AMPK)[37]. Sirt1 has been shown to interact with PGC-1α to regulate mitochondrial biogenesis and fatty acid oxidation in skeletal muscle[38,39]. AMPK in skeletal muscle directly phosphorylates PGC-1α, increasing fatty acid oxidation and mitochondrial biogenesis[40] and Sirt6 has been shown to regulate AMPK in skeletal muscle[21]. Our observations that both PGC-1α and AMPK activity are downregulated in *Sirt6* KO mice (Supplementary Fig. 18) suggest a possible, if partial, role for AMPK-PGC-1α in mediating the

regulatory effects of Sirt6 on mitochondrial biogenesis and muscle oxidative capacity.

The determination of myofiber types is dynamically regulated and coordinated by several transcriptional activators and repressors[41]. To identify the molecules responsible for mediating the effects of Sirt6 on muscle fiber-type specification, we first performed RNA-seq analyses, determining that CREB and its target genes are those most affected by *Sirt6* ablation. Further, GTEx analysis showed a strong correlation between Sirt6 and CREB in human skeletal muscle. CREB is activated by exercise in human skeletal muscle[42,43] and drives gene expression changes capable of metabolic adaptation and muscle performance[35]. In this study, CREB expression and its target genes responsible for the regulation of fiber types and mitochondrial function, including PGC-1α and Nr4a3/1, were substantially repressed in *Sirt6* KO mice. Our study shows that Sirt6 promotes CREB transcription by enhancing RNAPII recruitment to enhancers and promoters of *Creb1*, despite diminished histone acetylation. Although deacetylation of histone proteins by Sirt6 generally correlates with chromatin condensation and gene silencing[44], there is also evidence that Sirt6 can activate certain genes while mediating histone deacetylation. For example, Sirt6 acts as a nuclear factor erythroid 2-related factor 2 (Nrf2) coactivator to protect against oxidative stress in human mesenchymal stem cells, wherein Sirt6 was found to be in a protein complex with Nrf2 and RNAPII[45], in accordance with our results.

With respect to the counterregulatory targets of Sirt6 against fast-to-slow fiber-type shift, we observed that Sox6, a transcriptional repressor, was negatively regulated by Sirt6. While a previous study using MIN6 pancreatic beta cells indicated that CREB binds to *Sox6* promoter to inhibit gene transcription[32], we reveal here a role of the CREB-Sox6 network in myofiber transition toward the slow fiber type. Sox6 belongs to the evolutionarily conserved Sox family, which is highly expressed in skeletal muscle[13]. Specifically, Sox6 suppresses the expression of slow fiber-specific genes by directly binding to their enhancer or promoter regions, thus indirectly inhibiting several transcriptional activators such as TEA domain family member 1 and 4 (TEAD1/4)[46]. Skeletal muscle-specific *Sox6* KO mice exhibited an increased proportion of slow myofibers and mitochondrial activity, leading to improved muscular endurance[14]. In this study, the altered proportion of slow fibers and concomitant repression of running endurance in *Sirt6* KO mice were completely reversed by AAV-mediated *Sox6* knockdown in skeletal muscle. CREB mRNA level, however, remained unchanged by *Sox6* silencing (data not shown), suggesting that CREB acts at the upstream level of Sox6.

Because Sirt6 has been implicated as a key regulator of aging and aging-related diseases, the Sirt6-CREB-Sox6 axis is expected to have diverse pathophysiological roles. Consistent with previous reports[28–30], we observed a more dramatic decline in Sirt6 levels in the muscles of 20-week-old mice than in those of 4-week-old mice. CREB signaling was disturbed in the brain of aged mice, and has been considered as a therapeutic target for age-related

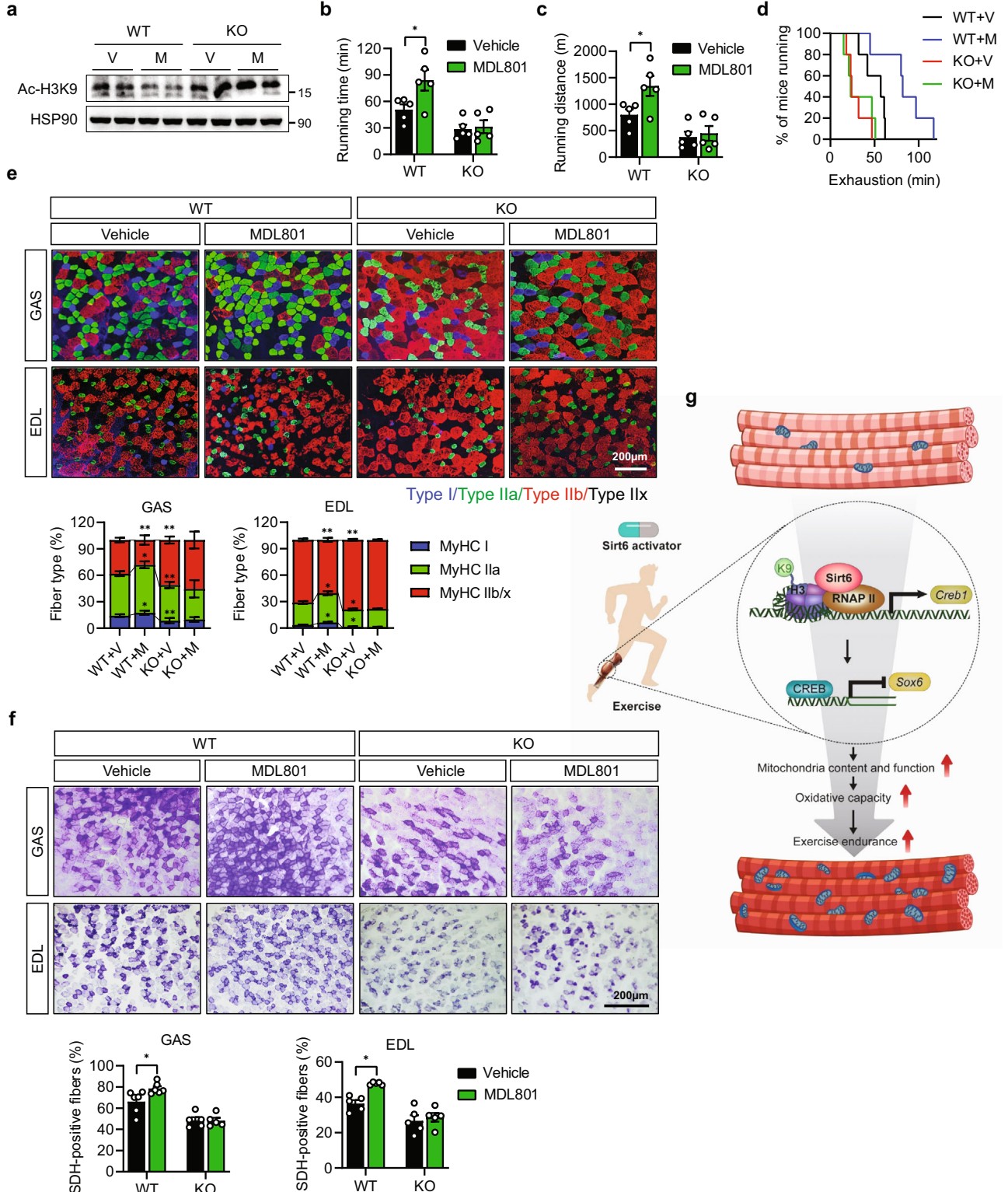

**Fig. 6 Increase in endurance exercise performance in mice treated with Sirt6 activator.** Twenty five-week old WT and *Sirt6* KO mice were treated with MDL801 (100 mg/kg) via oral gavage once a day for four weeks during treadmill exercise program. **a** Representative Western blot analysis of H3K9 acetylation in gastrocnemius muscles (GAS). **b**–**d** Running time, running distance, and time to exhaustion were compared between MDL801 (M)- and vehicle (V)-treated WT and *Sirt6* KO mice (*n* = 5). **e** Representative immunofluorescence staining for MyHC-I, MyHC-IIa and MyHC-IIb/x in GAS and extensor digitorum logus (EDL). Composition of each myofiber was quantified (*n* = 5). Bar=200 µm. **f** Representative succinate dehydrogenase (SDH) staining and quantification of SDH-positive fibers (*n* = 5). Bar = 200 µm. **g** Proposed model for Sirt6 regulation of myofiber composition. Values are mean ± SEM. Data are representative of at least three independent experiments. One-way ANOVA followed by Bonferroni's post hoc analysis (**e**) and two-way ANOVA followed by Tukey's post hoc analysis (**b**, **c**, **f**) were conducted for statistical analyses. *$p < 0.05$ and **$p < 0.01$. Source data are provided as a Source Data file.

cognitive deficits[47]. The Sirt6-CREB-Sox6 axis might accordingly be implicated in the treatment of various aging-related diseases as well as the study of muscle biology. Future studies to follow-up these results are well warranted.

Lastly, we determined that administrating the Sirt6 activator MDL801 to mice dramatically increased the proportion of slow fibers, mitochondrial activity, and exercise endurance with adequate regulation of the CREB-Sox6 pathway, confirming that Sirt6 activation leads to reprogramming of myofiber composition to adapt to endurance exercise. Despite much interest, the exercise mimetics that are currently known remain very limited: agonists of AMPK or PPARδ have been considered to have this effect[48]. In addition to the emerging value of Sirt6 as a therapeutic target for a variety of metabolic and inflammatory diseases[44], our findings add a further layer of clinical significance to the role of Sirt6 activators in enhancing exercise endurance.

## Methods

**Animal studies**. All animal experiments were performed in accordance with the Guide for the Care and Use of Laboratory Animals published by the US National Institutes of Health (NIH Publication No. 85-23, revised 2011). Sirt6flox/flox mice (Sirt6tm1.1Cxd/J, Stock No: 017334), Myl1/MLC1f-Cre mice (Myl1tm1(cre)sjb/J, Stock No: 024713), Sirt6 Tg mice (C57BL/6-Tg(RP23-352G18)1Coppa/J, Stock No: 028361), and R26R reporter mice (FVB.129S4(B6)-Gt(ROSA)26Sortm1Sor/J, Stock No: 009427) were obtained from the Jackson Laboratory (Bar Harbor, ME, USA). Sirt6flox/flox and Myl1-Cre mice were crossed to generate skeletal muscle-specific deletion of Sirt6, while hemizygous Sirt6 Tg mice were used for whole-body overexpression. Myl1-Cre and R26R reporter mice were crossed to visualize Cre-mediated recombination.

For pharmacological activation, MDL801 (Chemscene, Monmouth Junction, NJ, USA) dissolved in PBS containing 5% DMSO, 10% Titrisol and 30% Solutol was administered to mice orally at a dose of 100 mg/kg for 30 days. Mice had free access to food and water and were maintained in a room with controlled humidity (50%) and temperature (22 °C) on a 12-h light/dark cycle.

For Sox6 knockdown, twenty-week-old Sirt6 KO and age-matched wild-type male mice were intravenously injected with adeno-associated virus 9 (AAV9) scrambled control or AAV9-Sox6 shRNA (viral titer: $1 \times 10^{11}$ particles/mouse) (Genecopoeia, Rockville, MD, USA). Three weeks after AAV injection, muscle tissues (GAS) and heart were collected, and Sox6 expression was measured to confirm its knockdown. Mice were then subjected to treadmill running exercise.

For local overexpression of Sirt6 in skeletal muscles, control or AAV9-Sirt6 (Genecopoeia) was intramuscularly injected to mice (viral titer: $1 \times 10^{11}$ particles/mouse). At two weeks after AAV injection, mice were subjected to treadmill running exercise for 4 weeks.

All animal experiments were performed in accordance with the Guide for the Care and Use of Laboratory Animals published by the US National Institutes of Health (NIH Publication No. 85-23, revised 2011). The study protocol was approved by the Institutional Animal Care and Use Committee of Chonbuk National University (permit number: CBNU 2017-0087).

**Grip strength measurement and treadmill running**. Forelimb grip strength of mice (male, 25-week old) was measured using a digital grip-strength meter (Jeung Do Bio & Plant, Seoul, Korea) and normalized by lean body mass. A mouse grabbing a grid was horizontally pulled by its tail away from the grid and peak force was measured. Each mouse was tested three times, with each test taking place after a 15 min rest period. For treadmill running, a single-lane treadmill (Jeung Do Bio & Plant) was used. Mice were acclimatized to the treadmill with a daily 15 min run at 6 m/min for 5 days. Mice were then subjected to a daily chronic treadmill running test for 4 weeks. The treadmill ran at 8 m/min for the first 15 min followed by increases of 2 m/min every 10 min toward 16 m/min until exhaustion. Running time and distance were recorded for each mouse.

**Body fat and lean percentage**. The body fat percentage was determined using a Bruker Minispec mq 7.5 NMR analyzer (Bruker Optics, Ettlingen, Germany).

**Indirect calorimetry**. Mice (male, 25-week old) were housed in an Oxymax/CLAMS metabolic cage system from Columbus Instruments (Columbus, OH, USA) with one mouse/chamber. Mice were placed in metabolic cages for one day to adapt and avoid stress during analysis. After 24-h acclimatization, mice were monitored continuously for 72 h at ad libitum feeding in an environmental room set at 20–23 °C with 12 h–12 h (7:00 pm–7:00 am) dark-light cycles. The respiratory exchange ratio (VO$_2$/VCO$_2$) was measured by the Oxymax system. Data collected from the last 24 h of the experiment were used for analysis.

**Ex vivo muscle isometric force and fatigue measurement**. GAS muscles were dissected from the hind limbs of euthanized mice. Stainless steel hooks were tied to the tendons of the GAS muscles using 6-0 nylon sutures and the muscles were mounted vertically between a force transducer (Model 159901, Radnoti, Monrovia, CA, USA) and an adjustable hook. The muscles were immersed in an organ bath with platinum electrodes continuously perfused with O$_2$/CO$_2$ (95/5%)-saturated Krebs-Ringer solution (118 mM NaCl, 4.75 mM KCl, 24.8 mM NaHCO$_3$, 1.18 mM KH$_2$PO$_4$, 2.5 mM CaCl$_2$·2H$_2$0, 1.18 mM MgSO$_4$ and 10 mM glucose). At the start of each experiment, muscle length was adjusted to yield the maximum force (100 V for 2 ms) using a previously described protocol with slight modification[49]. After equilibration, GAS muscles were subjected to different stimulation frequencies to measure tetanic force (2 ms pulses at 10–200 Hz for 500 ms at 100 V with 1-min recovery intervals). Fatigue measurements of GAS muscles were evaluated through repeated bouts of stimulation, each lasting 7 min at a frequency of 1 Hz and 100 V. Data collection and analysis were performed using LabChart Pro Software (Version 8, ADInstruments, Colorado Springs, CO, USA). Muscle length and wet weight were measured at the end of each experiment.

**Histology**. Skeletal muscle tissues were immediately placed in 30% sucrose solution and embedded with liquid nitrogen-cooled isopentane. For SDH staining, frozen sections (10 μm) were incubated in 0.2 M sodium phosphate buffer solution (pH 7.6) containing 0.6 mM nitro blue tetrazolium and 50 mM sodium succinate (Sigma-Aldrich, St Louis, MO, USA) for 30 min at 37 °C. Slides were washed with DiH$_2$O and mounted with aqueous mounting media. For staining of myosin heavy chain isoforms, muscle sections were incubated overnight at 4 °C with primary antibodies for MyHC1 (BA-D5, 1:100 dilution, DSHB, lowa City, IA, USA), MyHC2a (SC-71, 1:100 dilution, DSHB), MyHC2b (BF-F3, 1:50 dilution, DSHB). After washing, secondary antibodies (Alexa Fluor 350-conjugated goat anti-mouse IgG2b, A21140, 1:100 dilution, Alexa Fluor 488-conjugated goat anti-mouse IgG1, A21121, 1:100 dilution, and Alexa Fluor 594-conjugated goat anti-mouse IgM, A21044, 1:100 dilution, Thermo Fisher Scientific, Waltham, MA, USA) were incubated for 1 h at 37 °C. Images were acquired using a Leica DM750 microscope (Leica, Wetzlar, Germany). Image analysis was performed using iSolution DT 36 software (Carl Zeiss, Oberkochen, Germany).

**Transmission electron microscopy (TEM) and image analysis**. For TEM analysis, GAS muscle samples were immediately fixed in 2% paraformaldehyde and 2% glutaraldehyde in 50 mM sodium cacodylate buffer (pH 7.4) overnight. The tissue samples were post-fixed for 1.5 h with 1% osmium tetroxide in 50 mM sodium cacodylate buffer, stained overnight with 0.5% uranyl acetate, and dehydrated in ethanol series. The samples were infiltrated with a mixture of propylene oxide and Epon 812 resin (EMS, Hatfield, PA, USA) for viewing and imaging under the Hitachi Bio-TEM (Tokyo, Japan) at the Chonbuk National University Electron Microscopy facility. At least five fields of view of intermyofibrillar and sub-sarcolemmal mitochondria populations were captured for each section. The total number of IMF and SS mitochondria with normal or abnormal morphology per 1 μm$^2$ was counted in each group. iSolution DT 36 software (Carl Zeiss) was used to calculate mitochondrial number.

**Cell culture, transient transfection and reporter gene assays**. C2C12 cells and HEK293T cells were obtained from ATCC (Manassas, VA, USA). C2C12 myoblasts were maintained in culture at <80% confluence in DMEM supplemented with 10% FBS. Differentiation of C2C12 cells was initiated by replacing 10% FBS by 2% horse serum (Gibco Life Technologies, Waltham, MA, USA). Differentiation media were changed every two days and cells at day 5 were considered as differentiated myotubes.

For knockdown of Creb1 or Sirt6, C2C12 cells were transfected with 50 μM of siRNAs targeting for Creb1 or Sirt6 (Bioneer, Daejeon, Korea) using Lipofectamine RNAiMAX Reagent (Thermo Fisher Scientific). Cells were maintained for 5 days in a differentiation medium containing 2% horse serum.

Creb1 promoter reporter activities were measured in HEK293T cells after Sirt6 overexpression or deletion. HEK293T cells were transfected with Creb1-GLuc-ON™ Promoter Reporter construct along with either control (Mock) or Sirt6 expressing plasmid, and incubated for 24 h. For Sirt6 KO studies, control cells and Sirt6 KO cells were transfected and incubated for 24 h with Creb1-GLuc-ON™ Promoter Reporter construct. Culture media were collected for luminance measurement and transcriptional response activity values were expressed as luminance fold changes. Creb1 promoter activity was measured using the Secrete-Pair Dual Luminescence Assay Kit (GeneCopoeia), which enables analysis of the activities of Gaussia Luciferase (GLuc) and Secreted Alkaline Phosphatase (SEAP) in cell culture media.

**Generation of Sirt6 KO cells using CRISPR/Cas9 system**. HEK293T cells lacking Sirt6 were generated using the CRISPR/Cas9 system. To generate the KO cell line, HEK293T cells were transfected using Lipofectamine 3000 (Invitrogen, Carlsbad, CA, USA) with 4 μg of all-in-one plasmid expressing Cas9, green fluorescent protein (GFP), and guide RNA (gRNA) reagent. After 24 h, GFP-expressing cells were sorted by FACS Aria III (BD, Franklin Lakes, NJ, USA) and were maintained with complete media. The KO cell line clone was validated by Western blotting of Sirt6. The gRNA sequences for human Sirt6 (NM_001193285.3) were 5′-acttgcccttatccgcgtaCGG-3′

and 5′-gctgatgccggcgcccgtgTGG-3′ targeting exon 1 and exon 2, respectively. The plasmid was designed and purchased from Sigma-Aldrich.

**Primary myofiber isolation**. For myofiber isolation, GAS muscles were enzymatically digested with 2% type 1 collagenase (Worthingon, Lakewood, NJ, USA) for 90 min in shaking water bath at 37 °C. Digested muscles were blocked in DMEM with 10% FBS. The single myofibers were released by applying gentle pressure to the muscle, washed with 2% horse serum several times and collected for Seahorse analysis.

**Mitochondrial respiration**. For Seahorse analysis (XF96, Agilent Technologies, Santa Clara, CA, USA), isolated myofibers and C2C12 myoblasts were seeded in XF24 plates. After six days of differentiation, C2C12 cells were treated overnight with vehicle (DMSO) or MDL801 (5 or 10 µM). One hour prior to beginning the assay, myofibers and C2C12 myotubes were changed to DMEM containing with 5 mM glucose and 1 mM pyruvate. Oxygen consumption rate (OCR) was then measured according to manufacturer instructions with the injection of the Seahorse XF Cell Mito Stress Test Kit (Agilent Technologies). Respiration was measured three times by injection of oligomycin (1 µM), FCCP (0.5 µM) and rotenone/antimycin A (1 µM). Data were normalized to protein content.

**Western blotting**. Tissue homogenates (20 µg) were separated by 10% SDS-PAGE and transferred to PVDF membranes. After blocking with 5% skim milk, blots were probed with primary antibodies against Sirt6 (#12486, 1:200 dilution, Cell Signaling Technology, Beverly, MA, USA), CREB (#9197, 1:1000 dilution, Cell Signaling Technology), p-CREB (#9198 1:1000 dilution, Cell Signaling Technology), pAMPKα (#50081, 1:1000 dilution, Cell Signaling Technology), AMPKα (#5831, 1:1000 dilution, Cell Signaling Technology), Sox6 (ab64946, 1:1000 dilution, Abcam, Cambridge, UK), T-OXPHOS (ab110413, 1:1000 dilution, Abcam), Mfn1 (ab57602, 1:1000 dilution, Abcam), OPA1 (612606, 1:1000 dilution, BD Biosciences, Franklin Lakes, NJ, USA), HSP90 (ADI-SPA-836-F, 1:1000 dilution, Enzo Life Sciences, Plymouth Meeting, PA, USA), PGC-1α (AB-3242, 1:1000 dilution, Millipore, Danvers, MA, USA), Ac-H3K9 (H9286, 1:1000 dilution, Sigma-Aldrich), α-myosin (M4276, 1:400 dilution, Sigma-Aldrich), Drp1 (sc-271583, 1:1000 dilution, Santa Cruz Biochemicals, Dallas, TX, USA), Fis1 (sc-376447, 1:1000 dilution, Santa Cruz Biochemicals), Nor1 (sc-393902, 1:1000 dilution, Santa Cruz Biochemicals), Nur77 (sc-365113, 1:1000 dilution, Santa Cruz Biochemicals), Sirt1 (sc-74504, 1:1000 dilution, Santa Cruz Biochemicals), HDAC11 (sc-390737, 1:1000 dilution, Santa Cruz Biochemicals), and Lamin B1 (sc-6216, 1:1000 dilution, Santa Cruz Biochemicals). For immunoprecipitation HEK293T cells were transfected with expression plasmids for Flag and Flag-Sirt6. After 24 h, 500 µg of protein precleared with protein G-agarose was incubated with anti-Sirt6 overnight at 4 °C, then with protein G-agarose at 4 °C for 2 h. Blots were probed with primary antibody against RNAPII (920101, 1:2500 dilution, BioLegend, San Diego, CA, USA), Sirt6 (#12486, 1:200 dilution, Cell Signaling Technology), and signals were detected with a Las-4000 imager (GE Healthcare Life Science, Pittsburgh, PA, USA).

**RNA isolation and real-time quantitative RT-PCR (qPCR)**. Total RNA was extracted from skeletal muscle tissues using TRIzol reagent (Invitrogen). First-strand cDNA was generated using the random hexamer primer provided in a first-strand cDNA synthesis kit (Applied Biosystems, Foster City, CA, USA). Specific primers for each gene (Supplementary Table 1) were designed using qPrimerDepot (http://mouseprimerdepot.nci.nih.gov). qPCR reactions were conducted in a final volume of 10 µl containing 10 ng of reverse-transcribed total RNA, 200 nM of forward and reverse primers and PCR master mix. qPCR was performed in 384-well plates using an ABI Prism 7900HT Sequence Detection System (Applied Biosystems). The mRNA level of each target gene of interest was normalized to that of *Gapdh* (in the case of nuclear-encoded genes) or 16 S rRNA (in case of mtDNA-encoded genes).

For mitochondrial DNA content analysis, total DNA was extracted using a genomic DNA purification kit (Qiagen, Hiaden, Germany). Relative mtDNA was quantified by qPCR using primers for the mitochondrially encoded gene cytochrome oxidase 2 (*Cox2*), normalized to the nuclear-encoded gene cyclophilin A (*Ppia*).

**Chromatin immunoprecipitation**. Muscle tissues (100 mg) were chopped and cross-linked by incubating cells in 1% formaldehyde for 15 min at room temperature. Cross-linking was arrested by 5 min of incubation with 125 mM glycine. ChIP assay was performed using ChIP Enzymatic Chromatin IP Kits (Cell Signaling Technology). Chromatins were immunoprecipitated overnight at 4 °C with antibodies to Sirt6 (#12486, 1:200 dilution, Cell Signaling Technology), nonspecific IgG (#2729, 1:250 dilution, Cell Signaling Technology), RNAPII (920101, 1:250 dilution, BioLegend), CREB (MA1-083, 1:250 dilution, Abcam), Ac-H3K9 (H9286, 1:250 dilution, Sigma-Aldrich), and Sox6 (ab64946, 1:250 dilution, Abcam). qPCR for the ChIP DNA was performed to determine an association between Sirt6, RNAPII, or Ac-H3K9 to the *Creb1* promoter, Sox6 to the *Myh7* promoter, or CREB to the *Sox6* promoter. Data were normalized to input. All primer sequences are listed in Supplementary Table 1.

**RNA sequencing (RNA-Seq) and data analysis**. RNA from GAS muscle of ~20-week-old male mice was used. A TruSeq RNA sample preparation Kit v2 (Illumina, San Diego, CA, USA, #RS-122-2001) was used to convert the poly-A containing mRNA in total RNA into a cDNA library using poly-T oligo-attached magnetic bead selection. Following mRNA purification, the RNA was physically fragmented prior to reverse transcription and cDNA generation. The fragmentation step resulted in an RNA-Seq library that included inserts ranging in size from approximately 100–400 bp. The average insert size in an Illumina TruSeq RNA sequencing library was approximately 200 bp. The cDNA fragments then underwent an end repair process, with the addition of a single 'A' base to the 3′ end followed by ligation of the adapters. The resulting products were then purified and enriched with PCR to create the final double-stranded cDNA library. Libraries were quantified using KAPA Library Quantification kits for Illumina Sequencing platforms according to the qPCR Quantification Protocol Guide (KAPA BIOSYSTEMS, KK4855), and qualified using TapeStation D1000 ScreenTape (Agilent Technologies). Indexed libraries were then submitted to an Illumina Hiseq 4000, and paired-end (2×100 bp) sequencing was performed by Macrogen (Seoul, Korea). The relative abundances of the gene were measured in Read Count using StringTie. Statistical analysis of differential gene expression was performed using abundance estimates for each gene in the samples. Genes with FPKM values greater than 1 in the samples were excluded. To facilitate $\log_2$ transformation, 1 was added to each FPKM value for filtered genes. Filtered data were $\log_2$-transformed and subjected to quantile normalization. The statistical significance of differential expression data was determined using fold change. Gene-enrichment analysis and KEGG pathway analysis for DEGs were also performed based on Gene Ontology (http://geneontology.org/) and KEGG pathway (https://www.genome.jp/kegg/) databases respectively.

**Bioinformatic analysis**. To examine expression levels of sirtuin family members in the skeletal muscles of sedentary and exercise-trained subjects, a public microarray dataset was analyzed (GSE9103). Precise information about characteristics of the subjects and exercise conditions were described in a previous report[50]. Briefly, transcript profiling was obtained using Vastus Lateralis skeletal muscle biopsy samples from ten sedentary and ten trained healthy young (18–30 years old) individuals. Sedentary individuals exercised less than 30 min per day, twice per week. Trained individuals had performed ≥1 h cycling or running 6 days per week over the past 4 years.

**Statistical analysis**. Data are expressed as the mean ± Standard Error of the Mean (SEM). Statistical comparisons among multiple groups were made using one-way analysis of variance (ANOVA) followed by Bonferroni's post hoc analysis and two-way ANOVA followed by Tukey's post hoc analysis. The significance of differences between two groups was determined using Student's unpaired $t$ test. Correlation coefficient was calculated between CRE8 and Sirt6. A $p$ value of less than 0.05 was considered significant. IBM SPSS version 27 were used for all statistical analysis.

**Reporting summary**. Further information on research design is available in the Nature Research Reporting Summary linked to this article.

## Data availability
The source data underlying all box plots, bar, and line graphs can be found in the online Source Data File as well as the original uncropped Western blots. Raw and processed RNA-seq datasets were deposited to NCBI's GEO database under the accession number (GSE186105). RNA-seq data can be accessed using the link. Public microarray data (GSE9103) can be accessed using the link. Source data are provided with this paper.

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

## Acknowledgements

This work was supported by a grant from the Medical Research Center Program (2017R1A5A2015061) and by grants from the Basic Science Research Program (2016R1D1A1B01015213, 2020R1A2C2004761, and 2021R1A2B5B02001462) through the National Research Foundation (NRF), which is funded by the Korean government (MSIP). The authors would like to thank the Writing Center at Chonbuk National University for its skilled proofreading service.

## Author contributions

B.H.P. and E.J.B. conceived the idea, designed the experiments, and wrote the manuscript. M.Y.S., C.Y.H., Y.J.M., and J.H.L. conducted the experiments and analyzed the data. B.H.P. and E.J.B. had primary responsibility for the final content. All authors read and approved the final manuscript.

## Competing interests

The authors declare no competing interests.
