## [Peer Review File · Nature Communications]

Reviewers' Comments:

Reviewer #1:

Remarks to the Author:

In this manuscript authors have investigated the role of SIRT6 in reprogramming of glycolytic and oxidative type of muscle fibers. Authors have used muscle-specific SIRT6 knockout mice and transgenic mice with whole body overexpression of SIRT6 to determine whether and how SIRT6 expression reprograms type of myofibers. They have also used an activator of SIRT6 (MDL801), *in vivo*, to study effect of pharmacological activation of SIRT6 on exercise endurance, myofiber type and mitochondrial functionality. The manuscript reports an important finding for the role Sirt6 in switching of fast muscle fibers in favor of oxidative fibers and a possible link of Sirt6-CREB-Sox6 axis of signaling in regulating this function.

Major Concerns:

- (1) A role Sox6 protein in downregulation of slow muscle fibers has been already shown (Ref 38). The novelty of this work is the demonstration of upstream steps (Sirt6 and CREB) regulating slow muscle fibers through suppression of Sox6 activity. In order to prove this axis of signaling, authors should also do experiments with CREB-KO cells, and demonstrate that the effect of Sirt6 in supporting slow muscle fibers lost when CREB is deleted.
- (2) Fig.4H: Authors have used SIRT6KO samples to test Sirt6 occupancy on Creb1 promoters/enhancers and show that it is reduced. It is obvious that in SIRT6KO cells there will not be much SIRT6 present in the sample to bind. Instead, authors should use some disease model or aging model (may be muscle-specific) to estimate SIRT6 occupancy on the Creb promoters.
- (3) Luciferase reporter assays are very haphazardly described in methods as wells in results. How SIRT6 knock-down was achieved in HEK293 cells? Was it by siRNA or by shRNA not described? How was SIRT6 over-expressed in the cells? Was the CREB promoter transfected or infected? All these details are lacking in the manuscript.
- (4) Because authors find change in mitochondrial contents as well as health of mitochondria, they should check mitochondrial fusion-fission proteins (Mfn1, Mfn2, OPA1, DLP/Drp1 and Fis1) to assess whether mitochondrial dynamics is affected. More so because at the mRNA expression level, of the five genes tested for mitochondrial oxidative phosphorylation by authors, only one gene (Cycs) shows significant change at mRNA level with SIRT6KD.
- (5) It is very hard to understand how authors have achieved specific muscle fiber type immunohistochemical staining (Fig 2, 5, 6) using all three primary antibodies from mouse on the same tissue sections. Isn't the specificity for the secondary antibody lost if all primary antibodies used are from the same species (in this case mouse)? All the three tagged-anti-mouse secondary antibodies can cross-react with all the three mouse primary antibodies.
- (6) How specific the AAV-associated Sox6 down-regulation to muscles or did other organs like heart show knockdown too? Sox6 is shown to be a multi-faceted transcription factor in different cell types and is involved in terminal differentiation of skeletal muscles by suppressing slow-fiber specific genes (PMID: 21985497). Global down-regulation Sox6 in mice may affect renal and cardiovascular system and contribute to some effects found in this study. Since Sox6 Knockout mouse is available, genetic approach may be considered to strengthen the observation.
- (7) Fig5G (Bar 4): Why there is no downregulation of Sox6 mRNA in WT sample with AAV-shSox6 injection? Also, need to provide a western blot figure to show downregulation of Sox6 at protein level in muscles/organs.
- (8) Similarly, upregulation of SIRT6 by MDL801 should also be shown at mRNA and protein level to confirm the effect was indeed through SIRT6. There is always a likelihood that some other sirtuin or off target molecules may be upregulated when one injects a pharmacological agent. H3K9 deacetylation can be occurring due to other lysine deacetylases like SIRT1 or HDAC11 which are not checked by the authors.
- (9) To further support the authors conclusion, effect of MDL801 should be examined in Sirt6KO and CREB KO cells.

Technical points related to presentation:

- (1) Provide the Jackson Lab catalogue number and explain the genotypes of the mice used in this study, especially Tg-SIRT6 and the R26R reporter mice. It should also be mentioned that Tg-SIRT6 mice used in this study were whole body SIRT6 over-expressing, and not muscle-specific. Also, provide age and sex of mice used in the study in the methods section.
- (2) There no experiment done with C2C12 cells in this study, however methodology section

mentions experiments using these cells at two places (Muscle cell culture and differentiation; Mitochondrial respiration). At the same time no methodology is provided for HEK293T transfections/infections.

(3) Since, there are multiple groups used to compare statistical significance in majority of experiments, it should be mentioned that unpaired t-test is used only where two groups are compared.

(4) Are you isolating mtDNA, using RNAiso Plus, a total RNA extraction reagent? Clarify it

(5) Pg5, line 74: Need to provide reference for the statement that Myl1-Cre is mainly active in post-mitotic type II myofibers.

(6) For RT-qPCR experiments, normalization to which housekeeping gene was carried out, not provided. While checking mitochondria-related gene expression, authors should prefer one of the mitochondrial genome-encoded gene expression to normalize with the "gene of interest" data. This would nullify the bias towards reduced mitochondrial contents.

(7) Pg6, line 109-110: Does the "switching of myofibers" mean same glycolytic fibers change their phenotype to oxidative type by changing their protein expression profile or the glycolytic fibers die/reduce in size and oxidative type increase in number and/or size to replace them?

(8) In Fig3F, though overall SIRT6KO shows reduced mitochondrial respiration compared to wild-type, oligomycin per se did not seem to inhibit ATP synthase, why?.

(9) Most of the places number (n) of animals or samples not mentioned. Even though the graphs are made with all the samples, it is necessary to mention n= used.

(10) In Fig 2G, the pair of bar graphs for Myh1 and for Tnnt3 expression look identical, but Tnnt3 is not significant while Myh1 is. This kind of calculations for significance are found throughout the manuscript and need to be revisited.

(11) Tabulate antibodies used in this study to provide catalog numbers and vendor (company).

(13) Was downregulation of Creb1 mRNA in SIRT6KO mice observed by RNAseq of GAS muscle too?

(14) For RER measurements, were the mice fasted for certain time or were they on standard chow for the 72-hour period of experiment? How was their food intake normalized?

(15) Fig4 I: This is a HEK293 cell experiment with knock-down of SIRT6, so indicate it as KD (not KO) and also the other WT bar should be "control" not wildtype (WT).

(16) Pg 8, line148: Why is it "surprising" that SIRT6 ablation increased H3K9 acetylation? Actually it is "fitting".

(17) Page 4 line, ref 13 is incorrectly cited.

Reviewer #2:

Remarks to the Author:

Review Summary

Skeletal muscle fiber type composition is a major determinant of exercise capacity and a key regulator of basal metabolic rate. In this study, the authors aimed to identify the role of SIRT6 in reprogramming muscle towards an oxidative fiber type composition. The authors found that muscle-specific deletion of Sirt6 decreased muscle oxidative capacity and led to a glycolytic-dominant fiber type composition. Conversely, transgenic overexpression of Sirt6 increased muscle oxidative capacity and led to an oxidative-dominant fiber type composition. Specifically, the authors demonstrated that SIRT6 functions to recruit a transcriptional complex to promote expression of CREB, which suppresses SOX6, a known transcriptional repressor of oxidative fiber-specific genes, thereby promoting an oxidative fiber type composition via SOX6 de-repression. The authors concluded that SIRT6 activity can be targeted to modulate myofiber type composition, muscle oxidative capacity, and exercise performance.

This manuscript provides mechanistic insights into the process of oxidative fiber type reprogramming by the SIRT6/CREB/SOX6 pathway. Overall, the experiments presented were well designed, and the results demonstrated are sufficient to support the main conclusion. However some major concerns need to be addressed.

Major Concerns

Tissue specificity

The role of SIRT6 in muscle as a positive regulator of muscle oxidative capacity and exercise has been demonstrated previously (Cui et al. 2017) using a cardiac and skeletal muscle-specific Mck-Cre model, without a clear effect on muscle mitochondrial biogenesis or fiber type composition. In this study, the authors adopted a skeletal muscle-specific Myl1-Cre model to specifically evaluate the effect of Sirt6 deletion in skeletal muscle while preserving its expression in the heart. Indeed, as shown in Supplementary Figure 1D, the expression level of Sirt6 in the heart is minimally affected compared to the levels in skeletal muscles, which are sufficiently reduced. This is presumably important, as maintaining cardiac function is essential for reliable assessment of physical performance in vivo. However, MYL1 expression is specific to type II fibers, and the deletion efficiency of SIRT6 is higher in glycolytic muscles than in oxidative muscles (e.g., EDL vs. SOL). Therefore, the spatial limitation of the Myl1-Cre system, especially for studying the regulation of myofiber type composition, needs to be discussed. On the other hand, the Sirt6BAC model for Sirt6 overexpression does not provide any tissue specificity. To improve specificity to muscle, the authors may consider using a ROSA26-Loxp-STOP-Loxp-Sirt6 system, which allows constitutive overexpression of Sirt6 in the presence of muscle-specific Cre.

Muscle development

Myosin light chain expression generally starts embryonically during the myoblast differentiation and fusion. In contrast, muscle creatine kinase expression generally begins perinatally during the later stage of myotube maturation and myofiber formation. The differential effects of Sirt6 deletion on mitochondrial biogenesis between the Mck-Cre and the Myl1-Cre models suggest a temporal effect of SIRT6 deletion, depending on the developmental stage of the modulated muscle. The authors use the term "myofiber switching" throughout the manuscript without carefully evaluating the developmental effect of the SIRT6 deletion. Therefore, it is unclear whether the changes in myofiber type composition result from developmental reprogramming towards a specific fiber type, or from "switching" as a result of conversion between different mature myofiber types. To better elucidate the mechanism of changes in myofiber composition, the authors may consider modulating the myofiber composition during the myogenesis of myoblasts, and assessing the myofiber conversion of mature myofibers in response to physiological stimulation.

Exercise response

Most experiments are presented with sufficient details, which are well designed and analyzed. However, some areas require additional details to ensure reproducibility. For example, the transcriptome of human skeletal muscle varies substantially, depending on the subject's characteristics (gender, age, fitness), mode of exercise (aerobic vs. resistance), and training status (acute exercise vs. exercise training). In some cases, the expression level of SIRT6 may change in response to exercise, but in most cases, it does not. The transcriptomic dataset GSE9103 is limited to physically active, lean, and healthy males in response to aerobic exercise training. However, even within this dataset, the effect of exercise training on SIRT6 is not clearly significant based on independent bioinformatic validation ($\log_{2}FC = -0.02$, $FDR = 0.71$, $n = 40$). It is particularly concerning that the details for the bioinformatic analysis for the GSE9103 human profile are not included in the manuscript. Moreover, if exercise does indeed induce SIRT6 expression in muscle, it is still unclear what signaling pathways lead to the induction. The authors may consider evaluating the signaling pathways associated with oxidative exercise training (e.g., excitation-transcription coupling, endocrine/paracrine signaling) to strengthen the mechanism of exercise-induced SIRT6 expression.

Metabolic assessment

It is concerning that the manuscript does not include any basic metabolic assessments of the Sirt6 mKO model. The statement "Sirt6 mKO had phenotypical normal gross appearance at birth" does not satisfactorily address this concern. Although the metabolic effects of SIRT6 in muscle have been previously characterized in the Mck-Cre model, it is essential to characterize the basic metabolic parameters in the present study with the Myl1-Cre model in order to interpret the

changes in skeletal muscle fiber type composition and oxidative capacity in their physiological context. For example, an increase in adiposity alone can substantially impact physical performance as assessed by endurance running due to a larger physical workload independent of muscle function. Therefore, without assessing body weight and adiposity, it is premature to attribute the physical performance deficit to impaired muscle oxidative capacity *in vivo*. Additionally, assessment of skeletal muscle function in isolation, such as *ex vivo* force frequency and fatigue, is essential to determine the functional effects of Sirt6 expression without the confounding physiological factors.

Minor Concerns

1. In addition to the changes in total SIRT6 levels (as shown in Figure 1B), data on the muscle SIRT6 activity (i.e., Ac-H3K9) in response to exercise training in mice should be presented.
2. Instead of normalizing grip strength to body weight (as shown in Figure 1C), grip strength results in Newton without normalization should be shown. Alternatively, dependent on the grip strength protocol, the specific force can be estimated by normalizing force to lean mass but not total body mass.
3. Please repeat the indirect calorimetry measurement (as shown in Figure 1G) and ensure animals are under minimal stress and have adequate food access. The baseline respiratory exchange ratio is expected to fluctuate in a diurnal pattern (as shown in Figure 1K), which is not observed in the control animals shown in Figure 1G.
4. Please confirm that the muscle fiber type composition, gene expression, and calorimetry (as shown in Figure 2 and Figure 3) were assessed at baseline levels and not from the same animals trained for endurance running (as shown in Figure 2 D-F, H-J).
5. Please quantify fiber size distribution in addition to fiber type composition to elucidate the fiber type shift dynamic. Based on the representative histological images, a fiber type composition shift away from type I fiber is associated with hypertrophy Type II x/b fibers as in the mKO (Figure 2 B, and Figure 5 I). On the other hand, a fiber type composition shift toward type I myofibers is associated with myofiber hypertrophy of Type IIa in the transgenic and drug-treated muscle (as shown in Figure F and Figure 6 C).
6. Please show muscle fiber type and size distribution across different muscles, especially in oxidative muscles (e.g., SOL) for KO and glycolytic muscle (e.g., EDL) for Tg and drug-treated. At the minimum, please show gene expression of key fiber type markers (e.g., Myh7 and Myh4).
7. The statement, "the results obtained from Sirt6 Tg mice also support the view that Sirt6 is required for myofiber switching to the oxidative phenotype" should be modified. Only loss-of-function approaches (as in the Sirt6 mKO) can test whether a factor is required for a process.
8. Please include quantifications for the "lower number of mitochondria" in addition to the representative EM images (as shown in Figure 3A). Figure 3B quantifies the proportion of normal vs. abnormal mitochondria, which does provide information on mitochondria number. Figure 3C quantifies the mtDNA/nDNA ratio, which is informative but is often influenced by mitochondrial dynamics (i.e., fission and fusion) especially given the mitochondria abnormalities. In addition, please correct the colored label for Figure 3B.
9. Please elaborate on the statement, "since transcriptional activators are counter-balanced by a number of transcriptional repressors in determining oxidative muscle fibers, we next measured expression of putative repressors in the Sirt6 KO mice" by providing additional context to improve readability. Have transcriptional activators been screened in the mKO mice?
10. Please include data on MDL801 treatment on Sirt6 mKO mice, in addition to the result from wildtype mice (as shown in Figure 6). The Sirt6 mKO is a critical control for testing the specificity of MDL801 on SIRT6 activation and is expected to negate its effects on myofiber type composition and muscle oxidative capacity.
11. Please expand the discussion on the role of the SIRT6/CREB/SOX6 pathway in the context of aging and aging-related diseases, as SIRT6 was introduced as an anti-aging histone deacetylase.
12. Please expand the discussion on the role of AMPK in the SIRT6/CREB/SOX6 pathway in modulating the muscle oxidative capacity, as AMPK, a key regulator of mitochondria biogenesis via PGC1 α , is known to be regulated by SIRT6 deletion and overexpression.
13. Please attach a Nature Reporting Summary as required by the journal for publication and is essential for evaluation of the study design and analysis.

REVIEWER COMMENTS

Reviewer #1 (Remarks to the Author):

In this manuscript authors have investigated the role of SIRT6 in reprogramming of glycolytic and oxidative type of muscle fibers. Authors have used muscle-specific SIRT6 knockout mice and transgenic mice with whole body overexpression of SIRT6 to determine whether and how SIRT6 expression reprograms type of myofibers. They have also used an activator of SIRT6 (MDL801), in vivo, to study effect of pharmacological activation of SIRT6 on exercise endurance, myofiber type and mitochondrial functionality. The manuscript reports an important finding for the role Sirt6 in switching of fast muscle fibers in favor of oxidative fibers and a possible link of Sirt6-CREB-Sox6 axis of signaling in regulating this function.

Major Concerns:

1. *A role Sox6 protein in downregulation of slow muscle fibers has been already shown (Ref 38). The novelty of this work is the demonstration of upstream steps (Sirt6 and CREB) regulating slow muscle fibers through suppression of Sox6 activity. In order to prove this axis of signaling, authors should also do experiments with CREB-KO cells, and demonstrate that the effect of Sirt6 in supporting slow muscle fibers lost when CREB is deleted.*

Response: In this comment, the Reviewer raises a question concerning the effect of Sirt6 in CREB-deficient cells. In response to this suggestion, we have conducted new experiments. Our results indicate that the ability of Sirt6 to induce myofiber type changes was abolished by CREB knockdown in C2C12 myotubes, affirming the importance of the Sirt6-CREB-Sox6 signaling axis in determining myofiber type composition. We described this point as follows;

In the Methods section,

For knockdown of *Creb1* or *Sirt6*, C2C12 cells were transfected with 50 pM of siRNAs targeting for *Creb1* or *Sirt6* (Bioneer, Daejeon, Korea) using Lipofectamine RNAiMAX Reagent (Thermo Fisher Scientific). Cells were maintained for 5 days in a differentiation medium containing 2% horse serum.

In the Results section,

To further demonstrate the role of the Sirt6-CREB axis as an upstream regulator for Sox6, we performed these experiments in CREB knockdown cells. The results indicated that the effect of Sirt6 in supporting slow muscle fiber was lost in the CREB knockdown C2C12 myotubes (Supplementary Figs. 15a and 15b).

Supplementary Figure 15. CREB silencing reversed Sirt6 mediated expression of slow

fiber genes. **a** Protein expression of the indicated genes was examined in C2C12 myotubes transfected with siRNA targeting negative control (NC) or CREB, which was followed by infection with Ad-LacZ or Ad-Sirt6. **b** The indicated genes were measured by qPCR (n=5-6). Values are mean \pm SEM. One-way ANOVA followed by Bonferroni's *post hoc* analysis was conducted for statistical analyses. *, $p < 0.05$ and **, $p < 0.01$. Source data are provided as a Source Data file.

Consistent with the observation in *Sirt6* Tg (Supplementary Fig. 12), MDL801 treatment increased the expression of *Creb1* and its target genes, while decreasing *Sox6* in mice (Supplementary Fig. 16d). In addition, the effects of MDL801 on the expression of myofiber type-specific genes were blocked by CREB silencing in C2C12 myotubes (Supplementary Fig. 16e).

Supplementary Figure 16. Increase in exercise performance in mice treated with Sirt6 activator. **a** -----. **d** qPCR analysis of *Creb1* and its target genes in gastrocnemius muscles of WT mice (n=4). **e** Expression of *Sox6* and myofiber specific genes was measured by qPCR in C2C12 myotubes treated with MDL801 (10 μ M) after transfection with negative control (siNC) or CREB siRNA (n=5). Values are mean \pm SEM. Unpaired two-tailed *t*-test between two groups (**b**, **d**) and one-way ANOVA followed by Bonferroni's *post hoc* analysis (**c**, **e**) were conducted for statistical analyses. *, $p < 0.05$ and **, $p < 0.01$. V, vehicle; M, MDL801. Source data are provided as a Source Data file.

2. Fig.4H: Authors have used *SIRT6KO* samples to test *Sirt6* occupancy on *Creb1* promoters/enhancers and show that it is reduced. It is obvious that in *SIRT6KO* cells there will not be much *SIRT6* present in the sample to bind. Instead, authors should use some disease model or aging model (may be muscle-specific) to estimate *SIRT6* occupancy on the *Creb* promoters.

Response: We have now performed additional experiments to confirm the association of *Sirt6* on the *Creb1* promoters in pathophysiological condition. The results obtained from the muscles of 4- and 20-week-old mice show that *Sirt6* protein levels and its ability to bind to the promoters and enhancers of *Creb1* were attenuated in 20-week-old mice relative to 4-week-old animals. We revised the Results as follows;

In the Results section,

Sirt6, as an anti-aging molecule, has previously been shown to be downregulated over ages in various tissues²⁸⁻³⁰, observations that were confirmed by our results that showed that *Sirt6* protein was markedly reduced in the muscles of 20-week-old mice relative to 4-week-old mice (Supplementary Fig. 13a). The association of *Sirt6* in *Creb1* promoters and enhancers

was significantly diminished in 20-week-old mice (Supplementary Fig. 13b), suggesting the Sirt6 regulation of CREB in pathophysiological conditions.

Supplementary Figure 13. Sirt6 protein level and its association on *Creb1* promoters and enhancers in muscles of 4- and 20-week-old mice at basal condition. **a** Sirt6 protein level was examined in gastrocnemius muscle of 4- and 20-week old mice. **b** Occupancy of Sirt6 on *Creb1* promoters or enhancers was evaluated by ChIP-qPCR assay in mice muscle in (a). Values are mean \pm SEM (n=4). One-way ANOVA followed by Bonferroni's *post hoc* analysis was conducted for statistical analyses. *, $p < 0.05$ and **, $p < 0.01$. Source data are provided as a Source Data file.

3. Luciferase reporter assays are very haphazardly described in methods as wells in results. How SIRT6 knock-down was achieved in HEK293 cells? Was it by siRNA or by shRNA not described? How was SIRT6 over-expressed in the cells? Was the CREB promoter transfected or infected? All these details are lacking in the manuscript.

Response: We apologize for our negligence in this regard. We revised the manuscript as follows to include more detailed information about experiments.

In the Methods section,

Cell culture, transient transfection and reporter gene assays

For knockdown of *Creb1* or *Sirt6*, C2C12 cells were transfected with 50 pM of siRNAs targeting for *Creb1* or *Sirt6* (Bioneer, Daejeon, Korea) using Lipofectamine RNAiMAX Reagent (Thermo Fisher Scientific). Cells were maintained for 5 days in a differentiation medium containing 2% horse serum.

Creb1 promoter reporter activities were measured in HEK293T cells after Sirt6 overexpression or deletion. HEK293T cells were transfected with *Creb1*-GLuc-ON™ Promoter Reporter construct along with either control (Mock) or Sirt6 expressing plasmid, and incubated for 24 h. For Sirt6 KO studies, control cells and Sirt6 KO cells were transfected and incubated for 24 h with *Creb1*-GLuc-ON™ Promoter Reporter construct. Culture media were collected for luminance measurement and transcriptional response activity values were expressed as luminance fold changes. *Creb1* promoter activity was measured using the Secrete-Pair Dual Luminescence Assay Kit (GeneCopoeia), which enables analysis of the activities of *Gaussia* Luciferase (GLuc) and Secreted Alkaline Phosphatase (SEAP) in cell culture media.

Generation of Sirt6 KO cells using CRISPR/Cas9 system

HEK293T cells lacking Sirt6 were generated using the CRISPR/Cas9 system. To generate the

KO cell line, HEK293T cells were transfected using Lipofectamine 3000 (Invitrogen, Carlsbad, CA, USA) with 4 μ g of all-in-one plasmid expressing Cas9, green fluorescent protein (GFP), and guide RNA (gRNA) reagent. After 24 h, GFP-expressing cells were sorted by FACS Aria III (BD, Franklin Lakes, NJ, USA) and were maintained with complete media. The KO cell line clone was validated by Western blotting of Sirt6. The gRNA sequences for human Sirt6 (NM_001193285.3) were 5'-acttgcccttatccgcgtaCGG-3' and 5'-gctgatgccggcgcccgtgTGG-3' targeting exon 1 and exon 2, respectively. The plasmid was designed and purchased from Sigma-Aldrich.

4. Because authors find change in mitochondrial contents as well as health of mitochondria, they should check mitochondrial fusion-fission proteins (Mfn1, Mfn2, OPA1, DLP/Drp1 and Fis1) to assess whether mitochondrial dynamics is affected. More so because at the mRNA expression level, of the five genes tested for mitochondrial oxidative phosphorylation by authors, only one gene (Cycs) shows significant change at mRNA level with SIRT6KD.

Response: We appreciate the Reviewer's comment. As suggested, we measured mRNA and protein levels of mitochondrial fusion-fission proteins in the skeletal muscles of Sirt6 KO and WT mice and found that gene expressions were not altered between the genotypes. We described this point as follows;

In the Results section,

Because mitochondrial content and quality control are affected by alterations in mitochondrial dynamics²⁷, we measured genes associated with mitochondrial fusion-fission proteins in the GAS muscles of Sirt6 KO and WT mice. In contrast with the marked suppression of genes related to mitochondrial biogenesis, mRNA and protein levels of mitochondrial fusion-fission genes (i.e., OPA1, Mfn1, Drp1, and Fis1) remained unaffected in KO relative to WT mice (Supplementary Figs. 11a and 11b), supporting the notion that the effect of Sirt6 on mitochondrial contents primarily arises as a result of the regulation of mitochondrial biogenesis rather than the control of dynamics.

Supplementary Figure 11. No effect of Sirt6 deficiency on mitochondrial fusion-fission gene expressions at basal condition. a, b The mRNA and protein levels of genes involved in mitochondrial dynamics were examined in gastrocnemius muscles of WT and Sirt6 KO mice. Values are mean \pm SEM (n=6). Unpaired two-tailed *t*-test between two groups was conducted for statistical analyses. Source data are provided as a Source Data file.

5. It is very hard to understand how authors have achieved specific muscle fiber type immuno-histochemical staining (Fig 2, 5, 6) using all three primary antibodies from mouse on the same tissue sections. Isn't the specificity for the secondary antibody lost if all primary antibodies used are from the same species (in this case mouse)? All the three tagged-anti-

mouse secondary antibodies can cross-react with all the three mouse primary antibodies.

Response: Because mice produce immunoglobulins (Ig) of various classes and subclasses (e.g., IgG1, IgG2, IgM) (Manning et al., 2012), multiple mouse monoclonal primary antibodies in a given experiment can have different subclass identities. By using subclass-specific secondary antibodies, different primary antibodies can be detected simultaneously in multiplex assays, even where they are from the same host species. A method using this principle was previously employed to specifically label muscle fiber subtypes (Wang et al., 2017). In our experiments, we also used subclass-specific secondary antibodies (Alexa Fluor 350-conjugated goat anti-mouse IgG2b; Alexa Fluor 488-conjugated goat anti-mouse IgG1; and Alexa Fluor 594-conjugated goat anti-mouse IgM) as mentioned in the Methods section of the original manuscript. To improve a reader's comprehension of this method, we have now included more precise information about the primary antibodies used in Table S2 of the revised manuscript.

Examples in another published papers:

Knudsen NH et al., Interleukin-13 drives metabolic conditioning of muscle to endurance exercise. *Science*. 2020;368(6490):eaat3987. doi: 10.1126/science.aat3987.

Manning CF, Bundros AM, & Trimmer JS. Benefits and pitfalls of secondary antibodies: Why choosing the right secondary is of primary importance. *PLoS One*, 2012;7(6). doi:10.1371/journal.pone.0038313

Wang C, Yue F, Kuang S. Muscle histology characterization using H&E staining and muscle fiber type classification using immunofluorescence staining. *Bio Protoc*. 2017;7(10):e2279. doi: 10.21769/BioProtoc.2279.

6. How specific the AAV-associated Sox6 down-regulation to muscles or did other organs like heart show knockdown too? Sox6 is shown to be a multi-faceted transcription factor in different cell types and is involved in terminal differentiation of skeletal muscles by suppressing slow-fiber specific genes (PMID: 21985497). Global down-regulation Sox6 in mice may affect renal and cardiovascular system and contribute to some effects found in this study. Since Sox6 Knockout mouse is available, genetic approach may be considered to strengthen the observation.

Response: The Reviewer has raised a concern regarding the tissue specificity of gene deletion mediated by AAV-Sox6 shRNA, and we appreciate the helpful comment. In this study, we used the AAV9 serotype for gene delivery to skeletal muscle. While AAV9 can be transduced to other tissues, including the heart, following intravenous administration, transduction to target skeletal muscles has been considered an effective strategy due to higher efficiency when compared to other tissues (e.g., heart and liver) (Katwal et al., 2013). To strengthen tissue specificity in the AAV9-shSox6 system, we examined Sox6 expression in

heart and kidney tissues, and also checked the functionality of these organs. The wet weights of the heart and kidney, their Sox6 protein levels, and the functional markers [i.e., plasma creatinine and blood urea nitrogen (BUN)] remained similar between groups of AAV9-scrambled and AAV9-shSox6, supporting the predominant depletion of Sox6 in skeletal muscle in our system. This data has been included and described in our revised manuscript as follows;

In the Results section,

The role of Sox6 in the downregulation of slow muscle fibers has been already identified¹⁴. To demonstrate the causal relationship of Sox6 to Sirt6-dependent muscle fiber type changes, and because the AAV9 serotype has been identified as an effective means of targeting skeletal muscle³³, we silenced *Sox6* in skeletal muscle using AAV9-mediated shSox6 delivery to mice (Fig. 5g). Western blotting, histology and biochemical analysis confirmed that the skeletal muscle-selective depletion of Sox6 had no effect on the heart and kidney tissues (Supplementary Figs. 14a-14e).

Supplementary Figure 14. Muscle specific knockdown of Sox6 with no effect on heart or renal function by AAV9-shSox6 delivery to mice. a, b Protein expression of Sox6 was examined in gastrocnemius muscle (a) and heart (b) of WT and *Sirt6* KO mice injected with AAV9-Scr or AAV9-shSox6. c, d The wet weight (c) and H&E staining (d) of heart (n=4-6).

e Plasma levels of blood urea nitrogen (BUN) and creatinine (n=5-6). Values are mean \pm SEM. One-way ANOVA followed by Bonferroni's *post hoc* analysis was conducted for statistical analyses. Source data are provided as a Source Data file.

Reference

30. Katwal AB, Konkalmatt PR, Piras BA *et al.* Adeno-associated virus serotype 9 efficiently targets ischemic skeletal muscle following systemic delivery. *Gene Ther* 2013; **20**:930-938.

7. Fig5G (Bar 4): Why there is no downregulation of Sox6 mRNA in WT sample with AAV-shSox6 injection? Also, need to provide a western blot figure to show downregulation of Sox6 at protein level in muscles/organs.

Response: In response to the Reviewer's comment, we have now examined the protein level of Sox6 in the skeletal muscles of mice administered AAV-shSox6 or AAV-Scr. Western blot results indicate that Sox6 protein expression was significantly lowered in both WT and KO mice after AAV-shSox6 injection. This information is now included in the revised supplementary Figure 14a.

Supplementary Figure 14. Muscle specific knockdown of Sox6 with no effect on heart or renal function by AAV9-shSox6 delivery to mice. a Protein expression of Sox6 was examined in gastrocnemius muscle of WT and *Sirt6* KO mice injected with AAV9-Scr or AAV9-shSox6.

8. Similarly, upregulation of *SIRT6* by MDL801 should also be shown at mRNA and protein level to confirm the effect was indeed through *SIRT6*. There is always a likelihood that some other sirtuin or off target molecules may be upregulated when one injects a pharmacological agent. H3K9 deacetylation can be occurring due to other lysine deacetylases like *SIRT1* or *HDAC11* which are not checked by the authors.

Response: MDL801 was identified as an allosteric activator of Sirt6, which means that it has no effect on Sirt6 expression itself. When we performed additional experiments to address these concerns raised by the Reviewer, we found that MDL801 treatment decreased the expression of Ac-H3K9 while having no effect on Sirt1 and HDAC11 protein expression, and that this effect was not demonstrated in Sirt6 KO cells. This data excludes the possibility of an off-target effect by MDL801. We described this point as follows;

In the Results section,

To provide a clinical proof-of-concept, we treated WT and *Sirt6* KO mice with the Sirt6 specific activator MDL801³⁴ for 4 weeks as they were subjected to chronic treadmill exercise. The on-target specific functionality of Sirt6 activator MDL801-as-a Sirt6-activator was confirmed by H3K9 deacetylation in the muscle tissues, which effect was completely gone in

Sirt6 KO muscles (Fig. 6a). Other deacetylases (*Sirt1* and HDAC11) targeting H3K9 were not altered by treatment with MDL801 (Supplementary Fig. 16a).

Figures 6 and S16a. Increase in endurance exercise performance in mice treated with *Sirt6* activator. Twenty five-week old WT and *Sirt6* KO mice were treated with MDL801 (100 mg/kg) via oral gavage once a day for four weeks during treadmill exercise program. **6a** Western blot of H3K9 acetylation in gastrocnemius muscles (GAS). **S16a** Western blot of *Sirt1* and HDAC11 in GAS. V, vehicle; M, MDL801.

9. To further support the authors conclusion, effect of MDL801 should be examined in *Sirt6*KO and CREB KO cells.

Response: Please see response #1 to Reviewer 1 and response #10 to Reviewer 2. In this comment the Reviewer is asking about the specificity of MDL801 for *Sirt6* and if CREB is necessary for its action. As requested, we treated MDL801 in *Sirt6*- or CREB-deficient conditions. The effect of MDL801 on the expression of myofiber type specific genes was blocked by CREB silencing in C2C12 myotubes. Moreover, the effects of MDL801 on muscle fiber type specification and exercise performance were successfully abolished in *Sirt6* KO mice. In sum, these results suggest that CREB is downstream of *Sirt6* but upstream of *Sox6*, and that MDL801 functioned in a *Sirt6* specific manner in regulating muscle fiber type specification. These results are described in the revised manuscript as follows;

In the Results section,

The on-target specific functionality of *Sirt6* activator MDL801 ~~as a *Sirt6* activator~~ was confirmed by H3K9 deacetylation in the muscle tissues, which effect was completely gone in *Sirt6* KO muscles (Fig. 6a). Other deacetylases (*Sirt1* and HDAC11) targeting H3K9 were not altered by treatment with MDL801 (Supplementary Fig. 16a). Consistent with the increase of RER (VCO_2/VO_2) and decrease of OCR caused by *Sirt6* deficiency (Figs. 1g and 3f), lower RER and higher OCR were observed in mice and C2C12 cells that received MDL801 treatment (Supplementary Figs. 16b and 16c), confirming the role of *Sirt6* activation in the shift towards oxidative phosphorylation. Notably, running time, running distance, and time to exhaustion had increased approximately 2-fold in MDL801-treated mice compared to vehicle (Figs. 6b-6d), an effect that was not observed in *Sirt6* KO mice. Consistent with the observation in *Sirt6* Tg (Supplementary Fig. 12), MDL801 treatment increased the expression of *Creb1* and its target genes, while decreasing *Sox6* in mice (Supplementary Fig. 16d). In addition, the effects of MDL801 on the expression of myofiber type-specific genes were blocked by CREB silencing in C2C12 myotubes (Supplementary Fig. 16e). In terms of functional consequences, we observed that mice treated with MDL801 showed increased slow fiber composition and size with a concomitant increase in mitochondrial oxidative capacity in their GAS and EDL muscles (Figs. ~~6e and 6d~~ 6e, 6f and Supplementary Fig. 17).

Figure 6. Increase in endurance exercise performance in mice treated with Sirt6 activator. Twenty five-week old WT and *Sirt6* KO mice were treated with MDL801 (100 mg/kg) via oral gavage once a day for four weeks during treadmill exercise program. **a** Western blot of H3K9 acetylation in gastrocnemius muscles (GAS). **b-d** Running time, running distance, and time to exhaustion were compared between MDL801 (M)- and vehicle (V)-treated WT and *Sirt6* KO mice (n=5). **(b)** qPCR analysis of muscle fiber type specification genes. **e** Representative immunofluorescence staining for MyHC-I, MyHC-IIa and MyHC-IIb/x in GAS and extensor digitorum logus (EDL). Composition of each myofiber was quantified (n=5). **f** Succinate dehydrogenase (SDH) staining and quantification of SDH-

positive fibers (n=5-6). **g Proposed model for Sirt6 regulation of myofiber composition.** Values are mean \pm SEM. Unpaired two-tailed *t*-test between two groups (**b**, **c**, **f**) and one-way ANOVA followed by Bonferroni's *post hoc* analysis (**e**) were conducted for statistical analyses. *, $p < 0.05$ and **, $p < 0.01$. Source data are provided as a Source Data file.

Supplementary Figure 16. Increase in exercise performance in mice treated with Sirt6 activator. a -----. **e** Expression of *Sox6* and myofiber specific genes was measured by qPCR in C2C12 myotubes treated with MDL801 (10 μ M) after transfection with negative control (siNC) or CREB siRNA (n=5). Values are mean \pm SEM. Unpaired two-tailed *t*-test between two groups (**b**, **d**) and one-way ANOVA followed by Bonferroni's *post hoc* analysis (**c**, **e**) were conducted for statistical analyses. *, $p < 0.05$ and **, $p < 0.01$. V, vehicle; M, MDL801. Source data are provided as a Source Data file.

Technical points related to presentation:

(1) Provide the Jackson Lab catalogue number and explain the genotypes of the mice used in this study, especially *Tg-SIRT6* and the R26R reporter mice. It should also be mentioned that *Tg-SIRT6* mice used in this study were whole body *SIRT6* over-expressing, and not muscle-specific. Also, provide age and sex of mice used in the study in the methods section.

Response: We provided these additional details and added information concerning whole body *Sirt6* overexpression in *Tg* mice in our revised manuscript.

In the Methods section,

Animal studies

All animal experiments were performed in accordance with the Guide for the Care and Use of Laboratory Animals published by the US National Institutes of Health (NIH Publication No. 85-23, revised 2011). *Sirt6*^{lox/lox} mice (B6;129-*Sirt6*^{tm1Ygu}/J, Stock No: 008041), *Myf1/MLC1f-Cre* mice (*Myf1*^{tm1(cre)sjb}/J, Stock No: 024713), *Sirt6* *Tg* mice (C57BL/6-Tg(RP23-352G18)1Coppa/J, Stock No: 028361), and R26R reporter mice (FVB.129S4(B6)-Gt(ROSA)26Sortm1Sor/J, Stock No: 009427) were obtained from the Jackson Laboratory (Bar Harbor, ME, USA). *Sirt6*^{lox/lox} and *Myf1-Cre* mice were crossed to generate skeletal muscle-specific deletion of *Sirt6*, while hemizygous *Sirt6* *Tg* mice were used for whole body overexpression. *Myf1-Cre* and R26R reporter mice were crossed to visualize Cre-mediated recombination.

Grip strength measurement and exercise endurance treadmill running

Forelimb grip strength of mice (male, 25-week old) was measured using a digital grip-strength meter (Jeung Do Bio & Plant, Seoul, Korea) and normalized by ~~body weight~~ lean body mass.

Indirect calorimetry

Mice (**male, 25-week old**) were housed in an Oxymax/CLAMS metabolic cage system from Columbus Instruments (Columbus, OH, USA) **with one mouse/chamber**.

RNA sequencing (RNA-Seq) and data analysis

RNA from GAS muscle of ~20-week-old **male** mice was used.

(2) There no experiment done with C2C12 cells in this study, however methodology section mentions experiments using these cells at two places (Muscle cell culture and differentiation; Mitochondrial respiration). At the same time no methodology is provided for HEK293T transfections/infections.

Response: We apologize for our negligence in this regard. During preparation of the revised manuscript, we conducted additional experiments using C2C12 myotubes. Information has now been added to our Methods section as appropriate. Our methodology for HEK293T cells was also included in the manuscript as follows;

In the Methods section,

Cell culture, transient transfection and reporter gene assays

For knockdown of *Creb1* or *Sirt6*, C2C12 cells were transfected with 50 pM of siRNAs targeting for *Creb1* or *Sirt6* (Bioneer, Daejeon, Korea) using Lipofectamine RNAiMAX Reagent (Thermo Fisher Scientific). Cells were maintained for 5 days in a differentiation medium containing 2% horse serum.

Creb1 promoter reporter activities were measured in HEK293T cells after *Sirt6* overexpression or deletion. HEK293T cells were transfected with *Creb1*-GLuc-ON™ Promoter Reporter construct along with either control (Mock) or *Sirt6* expressing plasmid, and incubated for 24 h. For *Sirt6* KO studies, control cells and *Sirt6* KO cells were transfected and incubated for 24 h with *Creb1*-GLuc-ON™ Promoter Reporter construct. Culture media were collected for luminance measurement and transcriptional response activity values were expressed as luminance fold changes. *Creb1* promoter activity was measured using the Secrete-Pair Dual Luminescence Assay Kit (GeneCopoeia), which enables analysis of the activities of *Gaussia* Luciferase (GLuc) and Secreted Alkaline Phosphatase (SEAP) in cell culture media.

Mitochondrial respiration

For Seahorse analysis (XF96, Agilent Technologies, Santa Clara, CA, USA), isolated myofibers and C2C12 myoblasts were seeded in XF24 plates. After six days of differentiation, C2C12 cells were treated overnight with vehicle (DMSO) or MDL801 (5 or 10 μ M). One hour prior to beginning the assay, myofibers and C2C12 myotubes were changed to DMEM containing with 5 mM glucose and 1 mM pyruvate. Oxygen consumption rate (OCR) was then measured according to manufacturer instructions with the injection of the Seahorse XF Cell Mito Stress Test Kit (Agilent Technologies). Respiration was measured three times by injection of oligomycin (1 μ M), FCCP (0.5 μ M) and rotenone/antimycin A (1 μ M). Data were normalized to protein content.

(3) Since, there are multiple groups used to compare statistical significance in majority of experiments, it should be mentioned that unpaired *t*-test is used only where two groups are compared.

Response: We thank the Reviewer for identifying this issue, which has now been corrected as follows;

In Methods section:

Data are expressed as the mean \pm Standard Error of the Mean (SEM). ~~Statistical comparisons were made using one-way analysis of variance followed by Fisher's *post hoc* analysis. The significance of differences between groups was determined using Student's unpaired *t* test. A *p* value of less than 0.05 was considered significant.~~ Statistical comparisons among multiple groups were made using one-way analysis of variance followed by Bonferroni's *post hoc* analysis. The significance of differences between two groups was determined using Student's unpaired *t*-test. A *p* value of less than 0.05 was considered significant. Correlation coefficient was calculated between CRE8 and Sirt6. A *p* value of less than 0.05 was considered significant. IBM SPSS version 27 were used for all statistical analysis.

(4) Are you isolating mtDNA, using RNAiso Plus, a total RNA extraction reagent? Calcify it.

Response: We apologize for our negligence in this regard. We used a genomic DNA extraction kit for isolation of mitochondrial DNA, which is now reflected in our revised manuscript as follows;

In the Methods section,

For mitochondrial DNA content analysis, total DNA was extracted using a ~~genomic DNA purification kit (Qiagen, Hiaden, Germany) RNAiso Plus (Takara, Tokyo, Japan).~~ Relative mtDNA was quantified by qPCR using primers for the mitochondrially encoded gene cytochrome oxidase 2 (*Cox2*), normalized to the nuclear-encoded gene cyclophilin A (*Ppia*).

(5) Pg5, line 74: Need to provide reference for the statement that *Myl1-Cre* is mainly active in post-mitotic type II myofibers.

Response: We have added appropriate references to this statement as follows;

In the Results section,

To further explore the physiological role of Sirt6 in skeletal muscle, we generated skeletal muscle-specific *Sirt6* KO mice by crossing *Sirt6*^{*lox/lox*} mice with mice expressing Cre recombinase under the control of the myosin light chain 1 promoter (*Myl1-Cre*), which is mainly active in postmitotic type II myofibers^{24,25} (Supplementary Fig. ~~S4a~~ 2a).

References

24. Choi S, Jeong HJ, Kim H *et al.* Skeletal muscle-specific Prmt1 deletion causes muscle atrophy via deregulation of the PRMT6-FOXO3 axis. *Autophagy* 2019; **15**:1069-1081.
25. Bi P, Yue F, Sato Y *et al.* Stage-specific effects of Notch activation during skeletal myogenesis. *Elife* 2016; **5**: e17355.

(6) For RT-qPCR experiments, normalization to which housekeeping gene was carried out, not provided. While checking mitochondria-related gene expression, authors should prefer one of the mitochondrial genome-encoded gene expression to normalize with the “gene of interest” data. This would nullify the bias towards reduced mitochondrial contents.

Response: Information concerning the housekeeping gene for RT-qPCR normalization was added to the revised manuscript as follows. Among the mitochondria-related genes examined in Fig. 3d, *Mtco1* and *Mtco2* are mtDNA-encoded genes.

In the Methods section,

RNA isolation and real-time quantitative RT-PCR (qPCR)

The mRNA level of each target gene of interest was normalized to that of *Gapdh* (in case of nuclear-encoded genes) or 16S rRNA (in case of mtDNA-encoded genes).

(7) Pg6, line 109-110: Does the “switching of myofibers” mean same glycolytic fibers change their phenotype to oxidative type by changing their protein expression profile or the glycolytic fibers die/reduce in size and oxidative type increase in number and/or size to replace them?

Response: Our findings demonstrate the role of Sirt6 in the CREB-Sox6 pathway for the regulation of myofiber type-specific gene expression, highlighting how it ultimately leads to changes to the size and proportion of oxidative and glycolytic fibers (Figs 2b, 2f, S5a, S5b, and S17). To clarify our description, we have rephrased the sentence as follows;

In the Results section,

~~The results obtained from *Sirt6* Tg mice also support the view that *Sirt6* expression is required for shift of skeletal muscle fiber type toward type I slow twitch oxidative myofiber (Figs. 2E-2H).~~ Overall, these findings suggest that Sirt6 mediates the shift of myofibers towards an oxidative phenotype by increasing the proportion of slow twitch myofibers.

Figures 2 and S5. Alteration of myofiber composition in gastrocnemius muscle of *Sirt6* KO and *Sirt6* Tg mice at basal condition. **2b, 2f** Immunofluorescence staining for MyHC-I, MyHC-IIa, and MyHC-IIb/x. Composition and cross-sectional area (CSA) of each myofiber were quantified (n=4-6). **S5a, S5b** On the basis of expression of MyHC-positive myofibers in Figure 2b (a) and Figure 2f (b), the cross-sectional area of each type was determined.

(8) In Fig3F, though overall *SIRT6*KO shows reduced mitochondrial respiration compared to wild-type, oligomycin per se did not seem to inhibit ATP synthase, why?.

Response: We repeated the experiments and replaced to the data, in which oligomycin worked well.

f

Figure 3. Decrease in oxidative capacity in *Sirt6* KO mice at basal condition. a -----. f Oxygen consumption rate (OCR) was measured using Seahorse XF analyzer in ~~isolated adult~~ myofibers ~~isolated~~ from GAS. Basal respiration, respiration related to ATP production (uncoupled, difference between OCR before and after oligomycin injection), and maximal respiration (difference between OCR after FCCP and antimycin A (AA)/rotenone injection) were determined (n=4-5).

(9) Most of the places number (n) of animals or samples not mentioned. Even though the graphs are made with all the samples, it is necessary to mention n= used.

Response: The numbers of animals or samples used has now been clearly described in all figure legends.

(10) In Fig 2G, the pair of bar graphs for *Myh1* and for *Tnnt3* expression look identical, but *Tnnt 3* is not significant while *Myh1* is. This kind of calculations for significance are found throughout the manuscript and need to be revisited.

Response: We appreciate the Reviewer’s meticulous reading of our manuscript and apologize for this mistake. We recalculated and corrected the statistical significance in Fig. 2g, and also checked and addressed any similar instances throughout the manuscript.

g

(11) Tabulate antibodies used in this study to provide catalog numbers and vendor (company).

Response: As suggested, we have now included antibody information in Supplementary Table 2.

(12) Was downregulation of *Creb1* mRNA in *SIRT6*KO mice observed by RNAseq of GAS muscle too?

Response: *Creb1* mRNA levels were also downregulated in the GAS muscles of *Sirt6* KO mice, which was included in the heatmap of the revised Fig. 4b.

(13) For RER measurements, were the mice fasted for certain time or were they on standard chow for the 72-hour period of experiment? How was their food intake normalized?

Response: The mice were given normal chow for the 72 h period of the experiment, which is a usual condition of RER measurement. Since RER is the ratio between metabolic production of CO₂ and consumption of O₂, food intake is not directly reflected in the value. How much time has elapsed since last food intake might affect RER at earlier time (Eur Respir J, 2015), but conditions were identical and food intake was comparable between the genotypes. We have described indirect calorimetry in additional detail as follows;

In the Methods section,

Indirect calorimetry

~~Mice were housed in an Oxymax/CLAMS metabolic cage system from Columbus Instruments (Columbus, OH, USA). The study was carried out continuously for 72 h, in an environmental room set at 20–23°C with 12 h–12 h (7:00 pm–7:00 am) dark-light cycles. Respiratory exchange ratio (VO₂/VCO₂) was measured by the Oxymax system. Data collected from day 3 of the experiment were used for analysis.~~ Mice (male, 25-week old) were housed in an Oxymax/CLAMS metabolic cage system from Columbus Instruments (Columbus, OH, USA) with one mouse/chamber. Mice were placed in metabolic cages for one day to adapt and avoid stress during analysis. After 24-h acclimatization, mice were monitored continuously for 72 h at ad libitum feeding in an environmental room set at 20–23°C with 12 h–12 h (7:00 pm–7:00 am) dark-light cycles. Respiratory exchange ratio (VO₂/VCO₂) was measured by the Oxymax system. Data collected from ~~day 3~~ the last 24 h of the experiment were used for analysis.

(14) Fig4 I: This is a HEK293 cell experiment with knock-down of SIRT6, so indicate it as KD (not KO) and also the other WT bar should be “control” not wildtype (WT).

Response: Consistent with major comment #3 and technical point #2, we apologize for the confusion. We used a CRISPR-Cas9 system to silence Sirt6 in HEK293T cells, which has now been clearly described in the revised Methods and Figure legends.

(15) Pg 8, line148: Why is it “surprising” that SIRT6 ablation increased H3K9 acetylation? Actually it is “fitting”.

Response: We apologize for the confusion. We have now corrected this sentence as follows;

In the Results section,

Consistent with the observed increase in H3K9 acetylation in *Sirt6* KO mice (Fig. 4g), Ac-H3K9 occupancy in the *Creb1* promoter had also increased (Fig. 4j) while recruitment of RNA polymerase II (RNAPII) to the *Creb1* promoter was significantly dampened (Fig. 4k). This suggests that the intriguing possibility of a deacetylase activity dependent recruitment of RNAP II by Sirt6. We additionally observed the direct interaction of Sirt6 and RNAPII by co-IP in HEK293 cells transfected with plasmid expressing *Sirt6* (Fig. 4l).

(16) Page 4 line, ref 13 is incorrectly cited.

Response: The references have now been correctly cited.

13 Hagiwara N, Yeh M, Liu A. Sox6 is required for normal fiber type differentiation of fetal skeletal muscle in mice. *Dev Dyn* 2007; **236**:2062-2076.

14 Quiat D, Voelker KA, Pei J *et al.* Concerted regulation of myofiber-specific gene expression and muscle performance by the transcriptional repressor Sox6. *Proc Natl Acad Sci U S A* 2011; **108**:10196-10201.

Reviewer #2 (Remarks to the Author):

Review Summary

Skeletal muscle fiber type composition is a major determinant of exercise capacity and a key regulator of basal metabolic rate. In this study, the authors aimed to identify the role of SIRT6 in reprogramming muscle towards an oxidative fiber type composition. The authors found that muscle-specific deletion of *Sirt6* decreased muscle oxidative capacity and led to a glycolytic-dominant fiber type composition. Conversely, transgenic overexpression of *Sirt6* increased muscle oxidative capacity and led to an oxidative-dominant fiber type composition. Specifically, the authors demonstrated that SIRT6 functions to recruit a transcriptional complex to promote expression of CREB, which suppresses SOX6, a known transcriptional repressor of oxidative fiber-specific genes, thereby promoting an oxidative fiber type composition via SOX6 de-repression. The authors concluded that SIRT6 activity can be targeted to modulate myofiber type composition, muscle oxidative capacity, and exercise performance.

This manuscript provides mechanistic insights into the process of oxidative fiber type reprogramming by the SIRT6/CREB/SOX6 pathway. Overall, the experiments presented were well designed, and the results demonstrated are sufficient to support the main conclusion. However some major concerns need to be addressed.

Major Concerns

(1) *Tissue specificity*

The role of SIRT6 in muscle as a positive regulator of muscle oxidative capacity and exercise has been demonstrated previously (Cui et al. 2017) using a cardiac and skeletal muscle-specific Mck-Cre model, without a clear effect on muscle mitochondrial biogenesis or fiber type composition. In this study, the authors adopted a skeletal muscle-specific Myl1-Cre model to specifically evaluate the effect of Sirt6 deletion in skeletal muscle while preserving its expression in the heart. Indeed, as shown in Supplementary Figure 1D, the expression level of Sirt6 in the heart is minimally affected compared to the levels in skeletal muscles, which are sufficiently reduced. This is presumably important, as maintaining cardiac function is essential for reliable assessment of physical performance in vivo. However, MYL1 expression is specific to type II fibers, and the deletion efficiency of SIRT6 is higher in glycolytic muscles than in oxidative muscles (e.g., EDL vs. SOL). Therefore, the spatial limitation of the Myl1-Cre system, especially for studying the regulation of myofiber type composition, needs to be discussed.

Response: We agree with the Reviewer's comments and concerns concerning the Myl1-Cre model used in our study. In response to the Reviewer's suggestion, we added information concerning the limitations of the current study to the Discussion section as follows;

We utilized multiple animal models and in vivo gene delivery tools to manipulate *Sirt6* gene expression specifically in skeletal muscle. Despite the varying degree of tissue specificity of *Sirt6* gene manipulation, the results of the gene deletion (i.e., skeletal muscle specific *Sirt6* KO (*Myl1-Cre*) as well as AAV-shRNA mediated gene deletion) and overexpression (i.e., global *Sirt6* Tg or intramuscular injection of AAV9-*Sirt6*) tests consistently affirmed that *Sirt6* is a vital driver of oxidative, slow-twitch myofibers. Interestingly, although *Myl1* expression is specific to type II fibers and this was the case in our study- i.e., deletion

efficiency of *Sirt6* was higher in glycolytic muscles than in oxidative muscles (e.g., EDL vs. SOL), the effect of *Sirt6* on muscle function and fiber type specification was demonstrated consistently across all muscle type (GAS, EDL and SOL). As we focused for simplicity's sake on the distinction between slow and fast myofibers in interpreting our initial experimental results, additional studies will be required to determine the slow fiber selective role of *Sirt6* and the degree to which *Sirt6*-CREB-Sox6 signaling participates in defining the complete spectrum of myofiber subtypes. In addition to the spatial limitation described above, our model *Myf1-Cre* mediated *Sirt6* KO mice raises the temporal limitation-developmentally early onset of *Sirt6* gene silencing-so it may be premature to draw a definitive conclusion the *Sirt6* enhances exercise performance by facilitating the fast-to-slow fiber shift.

On the other hand, the *Sirt6*BAC model for *Sirt6* overexpression does not provide any tissue specificity. To improve specificity to muscle, the authors may consider using a *ROSA26-Loxp-STOP-Loxp-Sirt6* system, which allows constitutive overexpression of *Sirt6* in the presence of muscle-specific *Cre*.

Response: We appreciate the Reviewer's helpful comment. To strengthen tissue specificity in our *Sirt6* overexpression model, but cognizant of time limits, we performed additional experiments with AAV9-mediated genes delivered through intramuscular injection rather than the mouse system suggested by the reviewer. Our results showed that local overexpression of *Sirt6* in skeletal muscle had the same effects on myofiber type changes and exercise performance as those observed in *Sirt6* global transgenic mice. These results have now been described as follows;

In the Results section,

Moreover, intramuscular overexpression of *Sirt6* using an AAV9 delivery system enhanced oxidative fiber type composition and exercise performance (Supplementary Figs. 8a-8e).

Supplementary Figure 8. Increase in oxidative fiber composition and exercise performance by muscular overexpression of *Sirt6*. **a** Overexpression of *Sirt6* was verified by Western blotting in gastrocnemius muscles of mice injected intramuscularly with control or AAV9-*Sirt6*. For muscular overexpression of *Sirt6*, control or AAV9-*Sirt6* was

intramuscularly injected to mice and after 2 weeks, mice were subjected to treadmill running exercise for 4 weeks. **b** Expression of markers of slow and fast fibers was compared by qPCR (n=6). **c-e** Average running time, distance, and time to exhaustion after four weeks of treadmill exercise training (n=7-8). Values are mean \pm SEM. Unpaired two-tailed *t*-test between two groups was conducted for statistical analyses. *, $p < 0.05$ and **, $p < 0.01$. Source data are provided as a Source Data file.

(2) Muscle development

Myosin light chain expression generally starts embryonically during the myoblast differentiation and fusion. In contrast, muscle creatine kinase expression generally begins perinatally during the later stage of myotube maturation and myofiber formation. The differential effects of Sirt6 deletion on mitochondrial biogenesis between the Mck-Cre and the Myl1-Cre models suggest a temporal effect of SIRT6 deletion, depending on the developmental stage of the modulated muscle. The authors use the term “myofiber switching” throughout the manuscript without carefully evaluating the developmental effect of the SIRT6 deletion. Therefore, it is unclear whether the changes in myofiber type composition result from developmental reprogramming towards a specific fiber type, or from “switching” as a result of conversion between different mature myofiber types. To better elucidate the mechanism of changes in myofiber composition, the authors may consider modulating the myofiber composition during the myogenesis of myoblasts, and assessing the myofiber conversion of mature myofibers in response to physiological stimulation.

Response: The Reviewer raises an excellent point, questioning whether the effect of Sirt6 deletion relates to developmental reprogramming or is the result of conversion between different mature myofiber types. To directly address this question, we performed new experiments that introduced Sirt6 siRNA in different stages of myogenesis to C2C12. The results showed that silencing Sirt6 induced changes in myofiber specific gene expression (i.e., a decrease in the number of slow fiber genes and an increase in the number of fast fiber genes) in C2C12 myotubes transfected with Sirt6 siRNA both at Day 1 and Day 5 after differentiation. These results indicate that Sirt6 may act as a regulator of muscle fiber type during myogenesis and in mature myofibers for conversion. Our tests and results have been described in our revised manuscript as follows;

In the Results section,

Since *Myl1-Cre* action starts embryonically during myoblast differentiation^{25,26}, we further inquired whether the effect of Sirt6 deletion on myofiber composition is a consequence of developmental reprogramming or arises as a result of conversion between different types of mature myofibers. We introduced Sirt6 siRNA into C2C12 cells at various stages of myogenesis. Regardless of transfection time (at day 1 or 5 after differentiation), Sirt6 silencing decreased the mRNA levels of slow fiber genes but increased the mRNA levels of fast fiber gene in C2C12 myotubes (Supplementary Figs. 9a and 9b). These results were verified in tests with electrical stimulation (Supplementary Figs. 9c and 9d). In sum, our results indicate that Sirt6 may act as a regulator of muscle fiber type composition during myogenesis and in mature myofibers for conversion.

Supplementary Figure 9. Effect of Sirt6 silencing on myofiber specific gene expressions in different stages of myogenesis. **a, b** Slow and fast fiber gene expressions were examined by qPCR in C2C12 myotubes transfected with negative control (NC) or Sirt6 siRNA after differentiation (at day 1, **a** or at day 5, **b**) (n=4-5). **c, d** The experiments were done as in panel **(a)** and **(b)** in combination with electrical stimulation (1 ms pulses at 1 Hz for 6 h at 10 V) (n=4-5). Values are mean \pm SEM. Unpaired two-tailed *t*-test between two groups (**a, b**) and one-way ANOVA followed by Bonferroni's *post hoc* analysis (**c, d**) were conducted for statistical analyses. *, $p < 0.05$ and **, $p < 0.01$. Source data are provided as a Source Data file.

In the Discussion section,

We demonstrated that mitochondrial biogenesis was significantly reduced in *Myf1-Cre Sirt6* KO mice, a result that conflicts with a previous report by Cui et al.²¹, in which *Mck-Cre* mediated *Sirt6* deletion did not show any change in mitochondrial biogenesis. The difference may be a result consequence of the timing of *Sirt6* deletion in the developmental stages. While myosin light chain (*Myf1*) expression starts embryonically (E9), muscle creatine kinase (*Mck*) expression generally begins perinatally during the later stage of myotube maturation and myofiber formation (E15)^{25,26}. Impairment of mitochondrial biogenesis in *Myf1-Cre Sirt6* KO mice may be the result of temporal effect of *Sirt6* deletion, depending on the developmental stage of the modulated muscle.

References

- 25 Bi, P. *et al.* Stage-specific effects of Notch activation during skeletal myogenesis. *Elife* **5**, e17355, doi:10.7554/eLife.17355 (2016).
- 26 Mourkioti, F., Slonimsky, E., Huth, M., Berno, V. & Rosenthal, N. Analysis of CRE-mediated recombination driven by myosin light chain 1/3 regulatory elements in embryonic and adult skeletal muscle: a tool to study fiber specification. *Genesis* **46**, 424-430, doi:10.1002/dvg.20419 (2008).

(3) *Exercise response*

Most experiments are presented with sufficient details, which are well designed and analyzed. However, some areas require additional details to ensure reproducibility. For example, the transcriptome of human skeletal muscle varies substantially, depending on the subject's characteristics (gender, age, fitness), mode of exercise (aerobic vs. resistance), and training status (acute exercise vs. exercise training). In some cases, the expression level of SIRT6 may change in response to exercise, but in most cases, it does not. The transcriptomic dataset GSE9103 is limited to physically active, lean, and healthy males in response to aerobic exercise training. However, even within this dataset, the effect of exercise training on SIRT6 is not clearly significant based on independent bioinformatic validation (logFC=-0.02, FDR=0.71, n=40). It is particularly concerning that the details for the bioinformatic analysis for the GSE9103 human profile are not included in the manuscript.

Response: We appreciate the Reviewer's helpful comments. As pointed out, the effects of exercise on *SIRT6* expression might vary depending on context and pathophysiological situation. We have included precise information concerning our analysis of the transcriptome dataset as follows;

In the Methods section,

Bioinformatic analysis

To examine expression levels of sirtuin family members in the skeletal muscles of sedentary and exercise-trained subjects, a public microarray dataset was analyzed (GSE9103). Precise information about characteristics of the subjects and exercise conditions were described in a previous report⁵⁰. Briefly, transcript profiling was obtained using Vastus Lateralis skeletal muscle biopsy samples from ten sedentary and ten trained healthy young (18–30 years old) individuals. Sedentary individuals exercised less than 30 min per day, twice per week. Trained individuals had performed ≥ 1 h cycling or running 6 days per week over the past 4 years. Among these, one subject from each sedentary or trained group was excluded from analysis due to failure of array quality control.

Reference

50. Lanza IR, Short DK, Short KR *et al.* Endurance exercise as a countermeasure for aging. *Diabetes* 2008; **57**:2933-2942.

Moreover, if exercise does indeed induce SIRT6 expression in muscle, it is still unclear what signaling pathways lead to the induction. The authors may consider evaluating the signaling pathways associated with oxidative exercise training (e.g., excitation-transcription coupling, endocrine/paracrine signaling) to strengthen the mechanism of exercise-induced SIRT6 expression.

Response: We performed several new experiments to identify the signaling pathways that lead to Sirt6 induction in skeletal muscle in exercise condition. The results showed that IL-6 (as a paracrine/autocrine stimulus) and electrical stimulation (as an excitation signal) induced Sirt6 expression in C2C12 myotubes. These results have now been described in our revised manuscript as follows;

In the Results section,

Sirt6 induction may be mediated by endocrine/paracrine signaling and/or excitation-transcription coupling. To explore this possibility, we tested whether Sirt6 expression is changed by the exercise associated myokine interleukin (IL)-6²³ and electrical muscle stimulation. Treatment of C2C12 myotubes with recombinant IL-6 (5-50 ng/ml) raised Sirt6 protein expression in a concentration dependent manner (Supplementary Fig. 1a). Electrical muscle stimulation also increased Sirt6 expression in mice (Supplementary Fig. 1b), suggesting that multiple stimuli may coordinately contribute to Sirt6 upregulation in exercise conditions.

Supplementary Figure 1. Induction of Sirt6 by IL-6 treatment or electrical stimulation in C2C12 cells and gastrocnemius muscle. **a** Expression of Sirt6 was measured in C2C12 myotubes treated with recombinant IL-6 at indicated concentrations for 24 h. **b** Expression of Sirt6 was compared in gastrocnemius (GAS) muscle of mice with or without electrical pulse stimulation (EPS).

Reference

23 Steensberg, A. *et al.* Production of interleukin-6 in contracting human skeletal muscles can account for the exercise-induced increase in plasma interleukin-6. *J Physiol* **529 Pt 1**, 237-242, doi:10.1111/j.1469-7793.2000.00237.x (2000).

(4) Metabolic assessment

It is concerning that the manuscript does not include any basic metabolic assessments of the Sirt6 mKO model. The statement “Sirt6 mKO had phenotypical normal gross appearance at birth” does not satisfactorily address this concern. Although the metabolic effects of SIRT6 in muscle have been previously characterized in the Mck-Cre model, it is essential to characterize the basic metabolic parameters in the present study with the Myl1-Cre model in order to interpret the changes in skeletal muscle fiber type composition and oxidative capacity in their physiological context. For example, an increase in adiposity alone can substantially impact physical performance as assessed by endurance running due to a larger physical workload independent of muscle function. Therefore, without assessing body weight and adiposity, it is premature to attribute the physical performance deficit to impaired muscle oxidative capacity in vivo. Additionally, assessment of skeletal muscle function in isolation, such as ex vivo force frequency and fatigue, is essential to determine the functional effects of Sirt6 expression without the confounding physiological factors.

Response: We understand the Reviewer’s concern and have now provided relevant metabolic parameters, including body weight, epididymal adipose tissue weight, fat and lean mass, and food intake amount of the mice, all of which remained similar between *Sirt6* KO and WT mice. In addition, *Sirt6* KO mice appeared more susceptible to muscle fatigue, a result that was consistent with the results of myofiber types as oxidative muscles are generally more

resistant to fatigue than glycolytic muscles. This data has been included in the revised manuscript as follows;

In the Results section,

Sirt6 KO mice had a phenotypically normal gross appearance at birth and showed similar basic metabolic parameters (i.e., body weight, lean body mass, epididymal adipose tissue weight, and amount of food intake) as their wild type (WT) littermates (Supplementary Figs. 3a-3d).

Supplementary Figure 3. Basic metabolic parameters of *Sirt6* KO mice. Body weight (a), fat and lean mass (b), epididymal white adipose tissue (eWAT) weight (c), and food intake (d) were compared in WT and skeletal muscle specific *Sirt6* KO mice (male, 20-week old, n=7-10). Values are mean ± SEM. Unpaired two-tailed *t*-test between two groups were conducted for statistical analyses. Source data are provided as a Source Data file.

To further evaluate the effect of *Sirt6* deficiency on muscle function, we performed an *ex vivo* study to measure isometric force and fatigue sensitivity. The results indicated that GAS muscles from *Sirt6* KO mice exhibit reduced tetanic contraction and are more susceptible to fatigue than those from WT mice, though there was no significant difference in the twitch contraction between genotypes (Supplementary Figs. 4a-4d).

Supplementary Figure 4. Impaired force production and increased muscle fatigue in gastrocnemius muscles from *Sirt6* KO mice. a Isometric force and fatigue measurements were done from isolated mouse gastrocnemius muscles of WT and *Sirt6* KO mice. After mounting gastrocnemius muscles on a force transducer, the twitch force was measured by electrically stimulating the muscle with a single electrical pulse (100 V for 1 ms). b, c The tetanic force–frequency relationships were determined by triggering contraction using incremental stimulation frequencies (1 ms pulses at 10-200 Hz for 500 ms at 100 V). d Fatigue index was measured at 1 Hz and 100 V by repeated stimuli for 7 min and expressed as a percentage of the initial contractile force (n=5-6). Values are mean ± SEM. Unpaired two-tailed *t*-test between two groups was conducted for statistical analyses. *, p<0.05. Source data are provided as a Source Data file.

Minor Concerns

1. In addition to the changes in total SIRT6 levels (as shown in Figure 1B), data on the muscle SIRT6 activity (i.e., Ac-H3K9) in response to exercise training in mice should be presented.

Response: As suggested, Figure 1b now reflects a decreased level of H3K9 acetylation as well as enhanced Sirt6 expression in muscles responding to exercise.

In the Results section,

This significant increase in Sirt6 protein was also confirmed in exercise-trained mice **and accompanied a decrease in H3K9 acetylation as an indicator of Sirt6 activity** (Fig. 1b).

Figure 1. Alteration of exercise performance in *Sirt6* KO and *Sirt6* Tg mice. a -----. b Expression of Sirt6 in gastrocnemius muscle was compared in sedentary or exercise-trained mice (n=4).

2. Instead of normalizing grip strength to body weight (as shown in Figure 1c), grip strength results in Newton without normalization should be shown. Alternatively, dependent on the grip strength protocol, the specific force can be estimated by normalizing force to lean mass but not total body mass.

Response: As suggested, we re-estimated grip strength by normalizing force to lean mass, and found similar pattern in the results. This has replaced the data previously in Fig. 1c.

Figure 1. Alteration of exercise performance in *Sirt6* KO and *Sirt6* Tg mice. a -----. c Forelimb grip strengths were measured at 15 min intervals **and normalized against lean body mass** (n=6-7).

3. Please repeat the indirect calorimetry measurement (as shown in Figure 1G) and ensure animals are under minimal stress and have adequate food access. The baseline respiratory exchange ratio is expected to fluctuate in a diurnal pattern (as shown in Figure 1K), which is

not observed in the control animals shown in Figure 1G.

Response: Mice were placed in metabolic cages for one day for adaptation and to avoid stress during analysis, a fact which has been precisely described in our revised Methods section. In addition, we repeated our experiment and obtained RER results with fluctuation in a diurnal pattern. This data has replaced the prior information in Fig. 1g.

In the Methods section,

Indirect calorimetry

~~Mice were housed in an Oxymax/CLAMS metabolic cage system from Columbus Instruments (Columbus, OH, USA). The study was carried out continuously for 72 h, in an environmental room set at 20–23°C with 12 h–12 h (7:00 pm–7:00 am) dark–light cycles. Respiratory exchange ratio (VO_2/VCO_2) was measured by the Oxymax system. Data collected from day 3 of the experiment were used for analysis.~~ Mice (male, 25-week old) were housed in an Oxymax/CLAMS metabolic cage system from Columbus Instruments (Columbus, OH, USA) with one mouse/chamber. Mice were placed in metabolic cages for one day to adapt and avoid stress during analysis. After 24-h acclimatization, mice were monitored continuously for 72 h at ad libitum feeding in an environmental room set at 20–23°C with 12 h–12 h (7:00 pm–7:00 am) dark–light cycles. Respiratory exchange ratio (VO_2/VCO_2) was measured by the Oxymax system. Data collected from ~~day 3~~ the last 24 h of the experiment were used for analysis.

In the Results section,

Figure 1. Alteration of exercise performance in *Sirt6* KO and *Sirt6* Tg mice. a -----g, k Twenty-four hour light and dark cycle respiratory energy expenditure (RER) were measured by indirect calorimetry (n=3).

4. Please confirm that the muscle fiber type composition, gene expression, and calorimetry (as shown in Figure 2 and Figure 3) were assessed at baseline levels and not from the same animals trained for endurance running (as shown in Figure 2 D-F, H-J).

Response: We confirm that the experiments for the measurement of fiber type composition, gene expression, and calorimetry were performed under basal condition. This is now clearly described in the revised figure legends.

Figure 2. ~~Oxidative fiber density~~ Alteration of myofiber composition in gastrocnemius muscle of *Sirt6* KO and *Sirt6* Tg mice at basal condition.

Figure 3. Decrease in oxidative capacity in *Sirt6* KO mice at basal condition.

Supplementary Figure 6. Decrease in oxidative fiber density in the extensor digitorum longus (EDL) and soleus muscles (SOL) of *Sirt6* KO mice at basal condition.

Supplementary Figure 7. Selective increase in oxidative fiber genes in extensor digitorum longus (EDL) muscle of *Sirt6* Tg mice at basal condition.

Supplementary Figure 10. Increase in mitochondrial biogenesis in gastrocnemius muscle of *Sirt6* Tg mice at basal condition.

Supplementary Figure 11. No effect of *Sirt6* deficiency on mitochondrial fusion-fission gene expressions at basal condition.

Supplementary Figure 12. Increased CREB pathway in *Sirt6* Tg mice at basal condition.

Supplementary Figure 13. *Sirt6* protein level and its association on *Creb1* promoters and enhances in muscles of 4- and 20-week-old mice at basal condition.

Supplementary Figure S18. Reduction of AMPK activity in *Sirt6* KO mice at basal condition.

5. Please quantify fiber size distribution in addition to fiber type composition to elucidate the fiber type shift dynamic. Based on the representative histological images, a fiber type composition shift away from type I fiber is associated with hypertrophy Type II x/b fibers as in the mKO (Figure 2 B, and Figure 5 I). On the other hand, a fiber type composition shift toward type I myofibers is associated with myofiber hypertrophy of Type IIa in the transgenic and drug-treated muscle (as shown in Figure F and Figure 6 C).

Response: We additionally analyzed and quantified fiber size distribution in our revised Figs. 2b, 2f, S5, and S17, and found that fiber size was inversely changed with slow fiber type composition in various animal models. We described these results as follows:

Muscle fiber-type assessment of GAS muscles by immunofluorescent staining for myosin heavy chain (MyHC) isoforms revealed that *Sirt6* ablation led to decreased oxidative fiber density and increased glycolytic fiber size in comparison with ~~that~~ those of WT mice (Fig. 2b and Supplementary Fig. 5a). In keeping with this, whereas the mRNA expression of slow fiber-specific genes such as MyHC-I (*Myh7*), MyHC-IIa (*Myh2*), troponin I (*Tnni1*), troponin C (*Tnnc1*), and troponin T1 (*Tnnt1*) were significantly downregulated, the fast fiber-specific gene MyHC-IIb/x (*Myh4*) was dramatically upregulated in GAS muscle from the *Sirt6* KO mice (Fig. 2c).

6. Please show muscle fiber type and size distribution across different muscles, especially in oxidative muscles (e.g., SOL) for KO and glycolytic muscle (e.g., EDL) for Tg and drug-treated. At the minimum, please show gene expression of key fiber type markers (e.g., *Myh7* and *Myh4*).

Response: As suggested, we additionally examined proportion and size distributions of muscle fibers and myofiber specific gene expressions in different skeletal muscles. Ultimately, we obtained similar results as we had with GAS. We described these results as follows;

In the Results section,

Similar observations were made in the case of EDL and SOL muscles of *Sirt6* KO mice with regard to the proportion of myofibers and the expression of fiber type-specific genes (a much weaker effect was observed in SOL muscles presumably due to the predominant action of *Myf1-Cre* in fast muscles) (Supplementary Figs. S2a-S2e 6a-6c), whereas the opposite results were obtained from GAS and EDL muscles from whole body *Sirt6* Tg mice (Figs. 2e-2h and Supplementary Figs. 5b and 7).

Supplementary Figure 6. Decrease in oxidative fiber density in the extensor digitorum longus (EDL) and soleus muscles (SOL) of *Sirt6* KO mice at basal condition. **a** Immunofluorescence staining for MyHC-I, MyHC-IIa, and MyHC-IIb/x. Composition of each myofiber was quantified (n=4-5). **b** Expression of markers of slow and fast fibers was compared by qPCR (n=4-6). **c** Succinate dehydrogenase (SDH) staining and quantification of SDH-positive fibers (n=5). Values are mean \pm SEM. Unpaired two-tailed *t*-test between two groups was conducted for statistical analyses. *, $p < 0.05$. Source data are provided as a Source Data file.

Supplementary Figure 7. Selective increase in oxidative fiber genes in extensor digitorum longus (EDL) muscle of *Sirt6* Tg mice at basal condition. Expression of markers of slow and fast fibers was compared by qPCR (n=4). Values are mean \pm SEM. Unpaired two-tailed *t*-test between two groups was conducted for statistical analyses. *, $p < 0.05$. Source data are provided as a Source Data file.

7. The statement, “the results obtained from *Sirt6* Tg mice also support the view that *Sirt6* is required for myofiber switching to the oxidative phenotype” should be modified. Only loss-of-function approaches (as in the *Sirt6* mKO) can test whether a factor is required for a process.

Response: Consistent with our response to technical point #7 of Reviewer #1, the sentence has been appropriately rephrased to reflect the comments of both Reviewers as follows;

In the Results section,

Similar observations were made in the case of EDL and SOL muscles of *Sirt6* KO mice with regard to the proportion of myofibers and the expression of fiber type-specific genes (a much weaker effect was observed in SOL muscles presumably due to the predominant action of *Myf1-Cre* in fast muscles) (Supplementary Figs. S2a-S2e 6a-6c), whereas the opposite results were obtained from GAS and EDL muscles from whole body *Sirt6* Tg mice (Figs. 2e-2h and Supplementary Figs. 5b and 7) . . . Overall, these findings suggest that *Sirt6* mediates the shift of myofibers towards an oxidative phenotype by increasing the proportion of type I slow twitch myofibers.

8. Please include quantifications for the “lower number of mitochondria” in addition to the representative EM images (as shown in Figure 3A). Figure 3B quantifies the proportion of normal vs. abnormal mitochondria, which does provide information on mitochondria number. Figure 3C quantifies the mtDNA/nDNA ratio, which is informative but is often influenced by

mitochondrial dynamics (i.e., fission and fusion) especially given the mitochondria abnormalities. In addition, please correct the colored label for Figure 3B.

Response: Consistent with the Reviewer's comments, Fig. 3b now reflects the number of normal or abnormal mitochondria, and the label is now corrected. We also measured mRNA and protein levels of mitochondrial fusion-fission proteins in the GAS muscles of Sirt6 KO and WT mice. In contrast with the marked suppression of genes related to mitochondrial biogenesis (Fig. 3d), mitochondrial fusion-fission genes (*Mfn1*, *OPA1*, *DLP/Drp1* and *Fis1*) remained unaffected in KO relative to WT (Supplementary Figs. 11a and 11b), supporting the notion that the effect of Sirt6 on mitochondrial contents arises as a result of mitochondrial biogenesis regulation rather than the control of dynamics.

Figure 3. Decrease in oxidative capacity in *Sirt6* KO mice at basal condition. **a** Transmission electron micrograph of gastrocnemius muscles from WT and *Sirt6* KO mice. **b** Numbers of mitochondria with normal or abnormal morphology were counted from images in (a). **c** Mitochondrial DNA (mtDNA) was quantified by qPCR using nuclear DNA (nDNA) as a standard ($n=4-5$). **d** qPCR analysis of genes related to mitochondrial biogenesis and oxidative phosphorylation (OxPhos) in GAS ($n=6$). Expression of each gene was normalized with housekeeping *Gapdh* whereas expression of mitochondrial genome-encoded genes *Mtco1* and *Mtco2* was normalized with *16S rRNA*.

Supplementary Figure 11. No effect of Sirt6 deficiency on mitochondrial fusion-fission gene expressions at basal condition. **a, b** The mRNA and protein levels of genes involved in mitochondrial dynamics were examined in gastrocnemius muscles of WT and *Sirt6* KO mice. Values are mean \pm SEM ($n=6$). Unpaired two-tailed *t*-test between two groups was conducted for statistical analyses. Source data are provided as a Source Data file.

9. Please elaborate on the statement, “since transcriptional activators are counter-balanced by a number of transcriptional repressors in determining oxidative muscle fibers, we next measured expression of putative repressors in the *Sirt6* KO mice” by providing additional

context to improve readability. Have transcriptional activators been screened in the mKO mice?

Response: Using RNA seq, qPCR, and Western blotting analyses, we identified several transcriptional activators of CREB responses for slow muscle type specification (Figs. 4a-4d). Additionally, we rephrased the relevant sentence in our revised manuscript as follows;

~~Since transcriptional activators are counter-balanced by a number of transcriptional repressors in determining oxidative muscle fibers~~ The balance between transcriptional activators and repressors is critical in determining the number of oxidative muscle fibers^{5,31}. Since coactivator PGC-1 α , which is downstream of CREB, was altered by Sirt6 deficiency (Figs. 4c and 4d), we ~~measured expression levels of putative~~ checked the transcription repressors ~~of oxidative fiber conversion~~ in *Sirt6* KO mice.

10. Please include data on MDL801 treatment on *Sirt6* mKO mice, in addition to the result from wildtype mice (as shown in Figure 6). The *Sirt6* mKO is a critical control for testing the specificity of MDL801 on SIRT6 activation and is expected to negate its effects on myofiber type composition and muscle oxidative capacity.

Response: Consistent with our response to major comment #8 of Reviewer #1 concerning the specificity of MDL801, we conducted additional experiments using *Sirt6* mKO mice, the results of which indicated that all phenotype changes exerted by MDL801 in terms of myofiber type composition and exercise performance were not observed in *Sirt6* mKO mice. This highlights the functionality of MDL801 as a Sirt6 specific activator. This data was included and described as follows;

To provide a clinical proof-of-concept, we treated WT and *Sirt6* KO mice with the Sirt6 specific activator MDL801³⁴ for 4 weeks as they were subjected to chronic treadmill exercise. The on-target specific functionality of Sirt6 activator MDL801 ~~as a Sirt6 activator~~ was confirmed by H3K9 deacetylation in the muscle tissues, which effect was completely gone in *Sirt6* KO muscles (Fig. 6a). Other deacetylases (Sirt1 and HDAC11) targeting H3K9 were not altered by treatment with MDL801 (Supplementary Fig. 16a).

Figures 6 and S16a. Increase in endurance exercise performance in mice treated with Sirt6 activator. Twenty five-week old WT and *Sirt6* KO mice were treated with MDL801 (100 mg/kg) via oral gavage once a day for four weeks during treadmill exercise program. **6a** Western blot of H3K9 acetylation in gastrocnemius muscles (GAS). **S15a** Western blot of Sirt1 and HDAC11 in GAS.

11. Please expand the discussion on the role of the SIRT6/CREB/SOX6 pathway in the context of aging and aging-related diseases, as SIRT6 was introduced as an anti-aging histone

deacetylase.

Response: As suggested, the potential role of the Sirt6/CREB/Sox6 axis was further discussed in the context of aging and aging-related diseases as follows;

In the Discussion section,

Because Sirt6 has been implicated as a key regulator of aging and aging-related diseases, the Sirt6-CREB-Sox6 axis is expected to have diverse pathophysiological roles. Consistent with previous reports²⁸⁻³⁰, we observed a more dramatic decline in Sirt6 levels in the muscles of 20-week-old mice than in those of 4-week-old mice. CREB signaling was disturbed in the brain of aged mice, and has been considered as a therapeutic target for age-related cognitive deficits⁴⁷. The Sirt6-CREB-Sox6 axis might accordingly be implicated in the treatment of various aging-related diseases as well as the study of muscle biology. Future studies to follow-up these results are well warranted.

References

28 Moon, Y. J. *et al.* Sirtuin 6 in preosteoclasts suppresses age- and estrogen deficiency-related bone loss by stabilizing estrogen receptor alpha. *Cell Death Differ* **26**, 2358-2370, doi:10.1038/s41418-019-0306-9 (2019).

29 Zhang, J. *et al.* Are sirtuins markers of ovarian aging? *Gene* **575**, 680-686, doi:10.1016/j.gene.2015.09.043 (2016).

30 Shen, X. *et al.* Age-dependent role of SIRT6 in jawbone via regulating senescence and autophagy of bone marrow stromal cells. *J Mol Histol* **51**, 67-76, doi:10.1007/s10735-020-09857-w (2020).

47 Yu, X. W., Oh, M. M. & Disterhoft, J. F. CREB, cellular excitability, and cognition: Implications for aging. *Behav Brain Res* **322**, 206-211, doi:10.1016/j.bbr.2016.07.042 (2017).

12. Please expand the discussion on the role of AMPK in the SIRT6/CREB/SOX6 pathway in modulating the muscle oxidative capacity, as AMPK, a key regulator of mitochondria biogenesis via PGC1 α , is known to be regulated by SIRT6 deletion and overexpression.

Response: We thank the Reviewer for his or her helpful comments. AMPK signaling was suggested as being responsible for the regulatory effects of Sirt6 on metabolic homeostasis and mitochondrial function. Levels of p-AMPK were consistently significantly reduced in the GAS muscles of *Sirt6* KO mice. The potential role of AMPK was further discussed as follows;

In the Discussion section,

Although the CREB-Sox6 pathway was identified as a downstream target of Sirt6 in supporting myofiber specification and mitochondrial biogenesis, we propose that PGC-1 α , a key player in mitochondrial biogenesis in skeletal muscle^{8,9}, is an additional link in mediating the benefits that Sirt6 provides to muscle. It was previously reported that PGC-1 α is regulated by several upstream molecules including Sirt6³⁶, Sirt1, and AMP-activated protein kinase (AMPK)³⁷. Sirt1 has been shown to interact with PGC-1 α to regulate mitochondrial biogenesis and fatty acid oxidation in skeletal muscle^{38,39}. AMPK in skeletal muscle directly phosphorylates PGC-1 α , increasing fatty acid oxidation and mitochondrial biogenesis⁴⁰ and Sirt6 has been shown to regulate AMPK in skeletal muscle²¹. Our observations that both PGC-1 α and AMPK activity are downregulated in *Sirt6* KO mice (Supplementary Fig. 18) suggest a possible, if partial, role for AMPK-PGC-1 α in mediating the regulatory effects of Sirt6 on mitochondrial biogenesis and muscle oxidative capacity.

Supplementary Figure S18. Reduction of AMPK activity in *Sirt6* KO mice at basal condition. The levels of total- and phosphorylated AMPK α were examined by Western blotting in gastrocnemius muscles of twenty-five-week-old WT or *Sirt6* KO mice (n=4-7). Values are mean \pm SEM. Unpaired two-tailed *t*-test between two groups was conducted for statistical analyses. *, $p < 0.05$. Source data are provided as a Source Data file.

13. Please attach a Nature Reporting Summary as required by the journal for publication and is essential for evaluation of the study design and analysis.

Response: We have attached a Nature Reporting Summary.

Reviewers' Comments:

Reviewer #1:

None

Reviewer #2:
Remarks to the Author:

The authors included additional results and discussion to address the concerns in our last comment. Overall, the authors did a thorough job for revision and the manuscript is much improved. However, There is only one minor but important concern remaining which was not addressed adequately.

To establish the translational context, the authors analyzed public dataset GSE9103 from NCBI and found SIRT6 was unregulated by exercise training in human (Figure 1A), with 9 subjects in each group. However, there are 10 subjects in each of the author-defined group in this dataset.

Based on the expression pattern, it appears that the samples GSM230407 (Sed) and GSM230418 (Ex) were removed from the analysis. These samples likely contributed to the intra-group variability, which will not produce a significant difference between the groups after false discovery adjustment (FDR = 0.7). Given that there was a significant difference based on unequal variant t.test without false discovery adjustment ($p < 0.046$), we asked the authors to provide more details of the bioinformatic analysis to understand how the authors have reached their conclusion.

In the rebuttal letter, the author explained that “among these, one subject from each sedentary or trained group was excluded from analysis due to failure of array quality control.” This is debatable as the coverage for these two samples are very much comparable to the rest:

Therefore, to faithfully exclude samples based on quality control, the authors must describe a predefined set of thresholds for the inclusion and exclusion criteria, which is still lacking in the revised version of the manuscript or rebuttal letter.

In the final manuscript revision, please provide a description for a predefined set of threshold for the inclusion and exclusion criteria for sample quality control.

REVIEWER COMMENTS

Reviewer #2 (Remarks to the Author):

The authors included additional results and discussion to address the concerns in our last comment. Overall, the authors did a thorough job for revision and the manuscript is much improved. However, there is only one minor but important concern remaining which was not addressed adequately.

To establish the translational context, the authors analyzed public dataset GSE9103 from NCBI and found SIRT6 was unregulated by exercise training in human (Figure 1A), with 9 subjects in each group. However, there are 10 subjects in each of the author-defined group in this dataset.

Based on the expression pattern, it appears that the samples GSM230407 (Sed) and GSM230418 (Ex) were removed from the analysis. These samples likely contributed to the intra-group variability, which will not produce a significant difference between the groups after false discovery adjustment (FDR = 0.7). Given that there was a significant difference based on unequal variant t.test without false discovery adjustment ($p < 0.046$), we asked the authors to provide more details of the bioinformatic analysis to understand how the authors have reached their conclusion.

In the rebuttal letter, the author explained that “among these, one subject from each sedentary or trained group was excluded from analysis due to failure of array quality control.” This is debatable as the coverage for these two samples are very much comparable to the rest:

Therefore, to faithfully exclude samples based on quality control, the authors must describe a predefined set of thresholds for the inclusion and exclusion criteria, which is still lacking in the revised version of the manuscript or rebuttal letter.

In the final manuscript revision, please provide a description for a predefined set of threshold for the inclusion and exclusion criteria for sample quality control.

Response: We are very grateful for the reviewer’s overall positive views on our revision. Also, we appreciate for the detailed reviews and helpful valuable comments, and apologize for the insufficient information regarding bioinformatic analysis and related criteria.

For statistical analysis, the significance of differences between sedentary and exercised groups was determined using Student’s unpaired *t*-test as the raw *p* value without false discovery adjustment.

In addition, the exclusion of the samples (i.e., one subject from each sedentary or trained group) in the previous analysis were done based on the described information in the GSE9103 dataset as attached below.

Status	Public on Nov 01, 2007
Title	Skeletal Muscle Transcript Profiles in Trained or Sedentary Young and Old Subjects
Organism	Homo sapiens
Experiment type	Expression profiling by array
Summary	Aging is associated with mitochondrial dysfunction and insulin resistance. We conducted a study to determine the role of long-term vigorous endurance exercise on age-related changes in insulin sensitivity and various indices of mitochondrial functions. Keywords: The effect of excises in young and old human subjects by transcription profiling
Overall design	Skeletal muscle transcript profiling was done using Vastus Lateralis muscle biopsy samples from 10 young sedentary (YS), 10 older sedentary (OS), 10 young trained (YT) and 10 older trained (OT) men and women. Note that YT2, YS1, and OT1 didn't pass the Quality Control Step of dChip (high array/single outliers). Sedentary subjects exercised less than 30 min/day, twice per week. Trained subjects performed ≥ 1 hour cycling or running 6 days/week over the past 4 years.

Please refer to “Note that YT2, YS1, and OT1 didn't pass the Quality Control Step of dChip (high array/single outliers).” Thus, the YT2 (young trained subject #2) and YS1 (young sedentary subject #1) were excluded.

However, we do not have detailed information about quality control step and related criteria since we did not perform the microarray experiment ourselves, therefore we re-analyzed it without excluding the samples according to the reasonable comment of the reviewer.

As a consequence, we obtained same conclusion (i.e., increase in Sirt6 expression by exercise) and modified the heatmap which was reflected in the revised figure 1A. The method section of the manuscript was also revised.

In the Figure:

Fig. 1

In the Method section:

Bioinformatic analysis

To examine expression levels of sirtuin family members in the skeletal muscles of sedentary and exercise-trained subjects, a public microarray dataset was analyzed (GSE9103). Precise information about characteristics of the subjects and exercise conditions were described in a previous report ⁵⁰. Briefly, transcript profiling was obtained using Vastus Lateralis skeletal muscle biopsy samples from ten sedentary and ten trained healthy young (18–30 years old) individuals. Sedentary individuals exercised less than 30 min per day, twice per week. Trained individuals had performed ≥ 1 h cycling or running 6 days per week over the past 4 years. ~~Among these, one subject from each sedentary or trained group was excluded from analysis due to failure of array quality control.~~

We hope that the explanations related to the bioinformatic analysis and the results of re-analysis will be satisfactory to the reviewer.

Reviewers' Comments:

Reviewer #2:

Remarks to the Author:

The authors have addressed all concerns